# Protein gradients on the nucleoid position the carbon-fixing organelles of cyanobacteria

Joshua S MacCready[1], Pusparanee Hakim[2], Eric J Young[3], Longhua Hu[4], Jian Liu[4], Katherine W Osteryoung[5], Anthony G Vecchiarelli[2]*, Daniel C Ducat[3,6]*

[1]Department of Microbiology and Molecular Genetics, Michigan State University, East Lansing, United States; [2]Department of Molecular, Cellular, and Developmental Biology, University of Michigan, Michigan, United States; [3]Department of Biochemistry, Michigan State University, East Lansing, United States; [4]Biochemistry and Biophysics Center, National Heart, Lung, and Blood Institute, National Institutes of Health, Bethesda, United States; [5]Department of Plant Biology, Michigan State University, East Lansing, United States; [6]MSU-DOE Plant Research Laboratory, Michigan State University, East Lansing, United States

**Abstract** Carboxysomes are protein-based bacterial organelles encapsulating key enzymes of the Calvin-Benson-Bassham cycle. Previous work has implicated a ParA-like protein (hereafter McdA) as important for spatially organizing carboxysomes along the longitudinal axis of the model cyanobacterium *Synechococcus elongatus* PCC 7942. Yet, how self-organization of McdA emerges and contributes to carboxysome positioning is unknown. Here, we identify a small protein, termed McdB that localizes to carboxysomes and drives emergent oscillatory patterning of McdA on the nucleoid. Our results demonstrate that McdB directly stimulates McdA ATPase activity and its release from DNA, driving carboxysome-dependent depletion of McdA locally on the nucleoid and promoting directed motion of carboxysomes towards increased concentrations of McdA. We propose that McdA and McdB are a previously unknown class of self-organizing proteins that utilize a Brownian-ratchet mechanism to position carboxysomes in cyanobacteria, rather than a cytoskeletal system. These results have broader implications for understanding spatial organization of protein mega-complexes and organelles in bacteria.
DOI: https://doi.org/10.7554/eLife.39723.001

*For correspondence:
ave@umich.edu (AGV);
ducatdan@msu.edu (DCD)

**Competing interests:** The authors declare that no competing interests exist.

## Introduction

Bacterial microcompartments (BMCs) are protein-based organelles that encapsulate specialized metabolic processes in ~20% of all sequenced bacteria (*Axen et al., 2014*). Over 23 different classes of BMC are broadly distributed across bacterial phyla, but perhaps the best characterized class are the large (~175 nm) carbon-fixing carboxysomes of cyanobacteria (reviewed here: *Rae et al., 2013*; *Kerfeld and Melnicki, 2016*). Carboxysomes encapsulate the key Calvin-Benson-Bassham cycle enzyme, ribulose-1,5-bisphosphate carboxylase/oxygenase (RuBisCO), and enhance the carbon fixation efficiency of this enzyme by creating a microenvironment that is enriched for $CO_2$. Specifically, carboxysomes consist of an outer 'shell' layer that encapsulates both RuBisCO and carbonic anhydrase: the localized conversion of bicarbonate to $CO_2$ near RuBisCO is thought to greatly reduce its oxygenase activity, thereby suppressing the energetically-costly process of photorespiration (*Rae et al., 2013*). In the model rod-shaped cyanobacterium *Synechococcus elongatus* PCC 7942 (hereafter *S. elongatus*), it has been shown that carboxysomes align along the longitudinal axis of the cell (*Savage et al., 2010*). This equidistant positioning is thought to promote equal inheritance

**eLife digest** Cyanobacteria are tiny organisms that can harness the energy of the sun to power their cells. Many of the tools required for this complex photosynthetic process are packaged into small compartments inside the cell, the carboxysomes. In *Synechococcus elongatus*, a cyanobacterium that is shaped like a rod, the carboxysomes are positioned at regular intervals along the length of the cell. This ensures that, when the bacterium splits itself in half to reproduce, both daughter cells have the same number of carboxysomes.

Researchers know that, in *S. elongatus*, a protein called McdA can oscillate from one end of the cell to the other. This protein is responsible for the carboxysomes being in the right place, and some scientists believe that it helps to create an internal skeleton that anchors and drags the compartments into position.

Here, MacCready et al. propose another mechanism and, by combining various approaches, identify a new partner for McdA. This protein, called McdB, is present on the carboxysomes. McdB also binds to McdA, which itself attaches to the nucleoid – the region in the cell that contains the DNA. McdB forces McdA to release itself from DNA, causing the protein to reposition itself along the nucleoid. Because McdB attaches to McdA, the carboxysomes then follow suit, constantly seeking the highest concentrations of McdA bound to nearby DNA. Instead of relying on a cellular skeleton, these two proteins can organize themselves on their own using the nucleoid as a scaffold; in turn, they distribute carboxysomes evenly along the length of a cell.

Plants also obtain their energy from the sun via photosynthesis, but they do not carry carboxysomes. Scientists have tried to introduce these compartments inside plant cells, hoping that it could generate crops with higher yields. Knowing how carboxysomes are organized so they can be passed down from one generation to the next could be important for these experiments.
DOI: https://doi.org/10.7554/eLife.39723.002

of carboxysomes to both daughters of a dividing cell, which has been shown to impact cyanobacterial fitness under ambient $CO_2$ (*Savage et al., 2010*; *Rae et al., 2013*; *Kerfeld and Melnicki, 2016*). Since cyanobacteria contribute to greater than 25% of global carbon-fixation, how carboxysome organization is maintained is of considerable ecological importance. However, the mechanisms used to position any BMC, including carboxysomes, has remained elusive.

The most comprehensive study of BMC positioning to date showed that a ParA-type ATPase (hereafter McdA - Maintenance of carboxysome distribution A) displayed oscillatory dynamics in *S. elongatus* and was required for proper positioning of carboxysomes (*Savage et al., 2010*). ParA family members have been most comprehensively studied as factors important for the segregation of genetic material; bacterial chromosomes and low-copy plasmids (reviewed here: *Baxter and Funnell, 2014*). Early hypotheses of ParA function favored a cytoskeletal model in which ParA formed filaments that self-assemble into a larger scaffold capable of segregating genetic cargo, analogous to a primitive mitotic spindle (*Ringgaard et al., 2009*; *Ptacin et al., 2010*). These models were supported in part by *in vitro* observations of fibrous, long bundled filaments of purified members of the ParA/MinD family of proteins (Reviewed here: *Vecchiarelli et al., 2012*). Such filament-based models, combined with the requirement of McdA for carboxysome positioning, led to the prevailing theory that carboxysomes (*Savage et al., 2010*) and other BMCs (*Parsons et al., 2010*) could be tethered to a cryptic cytoskeletal element that traverses the cell length and/or exerts positioning forces.

More recently, filament-based models of ParA have been challenged by experiments utilizing reconstituted cell-free systems (*Hwang et al., 2013*; *Vecchiarelli et al., 2013*; *Vecchiarelli et al., 2014*), super-resolution microscopy (*Le Gall et al., 2016*; *Lim et al., 2014*), crystallography (*Zhang and Schumacher, 2017*), and mathematical modelling approaches (*Hu et al., 2015*; *Hu et al., 2017*; *Le Gall et al., 2016*; *Lim et al., 2014*; *Surovtsev et al., 2016*). These new data are consistent with a model whereby asymmetric distributions of ParA dimers on the nucleoid drive directed and persistent movement of DNA cargos (e.g. plasmids) towards increased concentrations of ParA via a Brownian-ratchet mechanism that does not require a cytoskeletal element. ParA-mediated DNA segregation via this proposed mechanism requires only three factors: (i) a ParA-type

ATPase that dimerizes and non-specifically binds the nucleoid in the presence of ATP (*Leonard et al., 2005*; *Hester and Lutkenhaus, 2007*; *Castaing et al., 2008*; *Vecchiarelli et al., 2010*), (ii) a partner protein, ParB, that site-specifically binds DNA and stimulates the ATPase activity of ParA to displace it from the nucleoid (*Davis et al., 1992*; *Bouet and Funnell, 1999*; *Bouet et al., 2000*), and (iii) a centromere-like site on the DNA cargo (*parS*) that ParB loads onto (*Davis and Austin, 1988*; *Funnell, 1988*). In this model, multiple dimers of ParB form a large protein-DNA complex around the *parS* site, which leads to a break in ParA symmetry across the nucleoid due to the formation of local ParA depletion zones around individual ParB-bound cargos (*Adachi et al., 2006*; *Hatano et al., 2007*; *Hwang et al., 2013*; *Vecchiarelli et al., 2014*). In turn, transient ParA-ParB interactions could translate the asymmetrical distribution of ParA across the nucleoid into a directional cue for processive motion of cargo towards the highest local concentration of ParA.

Despite extensive research into carboxysome biogenesis and organization, as well as efforts to identify the full complement of proteins associated with many BMC classes, the only evidence for an underlying cytoskeletal-mode of BMC organization is indirect. Brownian ratchet-based models are gaining favor within the ParA field, but there is not yet a broad consensus and there is substantial uncertainty as to whether ParA-type ATPases form functional filaments *in vivo* (*Wagstaff and Löwe, 2018*). Additionally, no factors suitable to play a role analogous to ParB or *parS* have been identified for McdA, preventing any direct assessment of a putative Brownian ratchet-based mechanism for carboxysomal organization. Therefore, whether McdA acts as part of a ParA-like system or utilizes a unique mechanism has remained an open question. Indeed, several fundamental questions remain unanswered in relation to carboxysome positioning, including: (i) Does McdA form a cytoskeletal structure or follow a Brownian-ratchet mechanism? (ii) Upon what cellular surface does McdA bind in order to processively oscillate from pole-to-pole? (iii) What factors contribute to the emergence of higher-order McdA patterning? (iv) How do oscillations of McdA contribute to carboxysome distribution?

Here, we identify a novel factor, we term McdB (Maintenance of carboxysome distribution protein B) that localizes to carboxysomes via interaction with outer shell proteins and regulates carboxysome positioning within the cytosol. While McdB has no identifiable similarities with any known ParB-family members, we find that McdB can directly interact with McdA to stimulate its ATPase activity and release it from DNA *in vitro*, and promote its pole-to-pole oscillation *in vivo*. Changes in McdB expression resulted in loss of McdA oscillatory dynamics, loss of equidistant carboxysome positioning and alteration of carboxysome ultrastructure. Although several features of McdAB differ significantly from those of classic ParAB family members, we find that a Brownian ratchet model of localized concentration gradients of McdA on the nucleoid is consistent with our results and may also reconcile past observations of carboxysome positioning. We discuss our results in light of their implications for BMC positioning and biogenesis, as well as the insights that this carboxysome positioning system can provide for a broader class of self-organizing proteins including the ParAB family.

## Results

### McdA dynamically patterns along the nucleoid

In the DNA partition process, ParA-type ATPases successively bind ATP, dimerize, and bind non-specifically to DNA (*Leonard et al., 2005*; *Hester and Lutkenhaus, 2007*; *Castaing et al., 2008*; *Vecchiarelli et al., 2010*). *In vivo*, this mechanism establishes the nucleoid as the biological surface upon which directed DNA cargo motion occurs (*Hatano and Niki, 2010*; *Sengupta et al., 2010*; *Castaing et al., 2008*; *Le Gall et al., 2016*). In the model rod-shaped cyanobacterium *S. elongatus*, the ParA-like protein we call McdA (Synpcc7942_1833; Maintenance of carboxysome distribution protein A) is required for positioning carboxysomes via an unknown oscillatory mechanism (*Savage et al., 2010*). However, the ParA family of ATPases contains MinD proteins, which bind membranes, ParA proteins, which bind DNA, and McdA has been broadly hypothesized to form filaments. Therefore, it remained unclear if McdA uses a biological surface to self-organize in the cell or if it forms a free-standing cytoskeletal network as previously proposed (*Savage et al., 2010*).

Since a C-terminal GFP fusion of McdA was previously used to observe McdA oscillation and its involvement in carboxysome positioning *in vivo* (*Savage et al., 2010*), we purified McdA-GFP-His and examined its capacity to bind DNA via an Electrophoretic Mobility Shift Assay (EMSA). We

observed that McdA-GFP-His significantly shifted non-specific DNA (nsDNA) in the presence of ATP (*Figure 1A*). To more directly examine the interaction of McdA with DNA, we used Total Internal Reflection Fluorescence Microscopy (TIRFM) to visualize McdA-GFP-His dynamics upon a DNA-coated surface; a technique that also has sufficient resolution to resolve oligomeric McdA filaments proposed to be involved in carboxysome positioning (*Savage et al., 2010*; *Yokoo et al., 2015*). A flowcell unit was decorated with nsDNA fragments (~500 bp sonicated salmon sperm DNA) at high density (~1000 fragments/μm$^2$) (*Vecchiarelli et al., 2013*) to create a visualizable DNA-coated surface. Consistent with EMSA analysis, McdA-GFP-His uniformly bound the DNA carpet when infused into the flowcell with ATP, (*Figure 1B* and *Video 1*), but showed no appreciable DNA binding when ADP or ATP-γ-S were added, or when nucleotides were omitted (*Figure 1C*). No McdA filaments were observed forming under any of these conditions (*Video 1*).

We then sought to examine endogenous localization and dynamics of McdA, therefore we generated N- and C-terminal fluorescent fusions of mNeonGreen (mNG) to McdA using the native *mcdA* promoter and chromosomal location. Interestingly, our native C-terminally tagged reporter (McdA-mNG) did not show dynamic oscillations and instead formed a uniform distribution of signal along the longitudinal axis (≥99% of cells; n = 950 cells) (*Figure 1D*, *Figure 1—figure supplement 1A* and *Table 1.*). Alternatively, an N-terminally tagged reporter (mNG-McdA) displayed robust oscillations (≥99% of cells; n = 442 cells) (periodicity of 15.3 min per 3.3 μm, ~5–6 x faster than

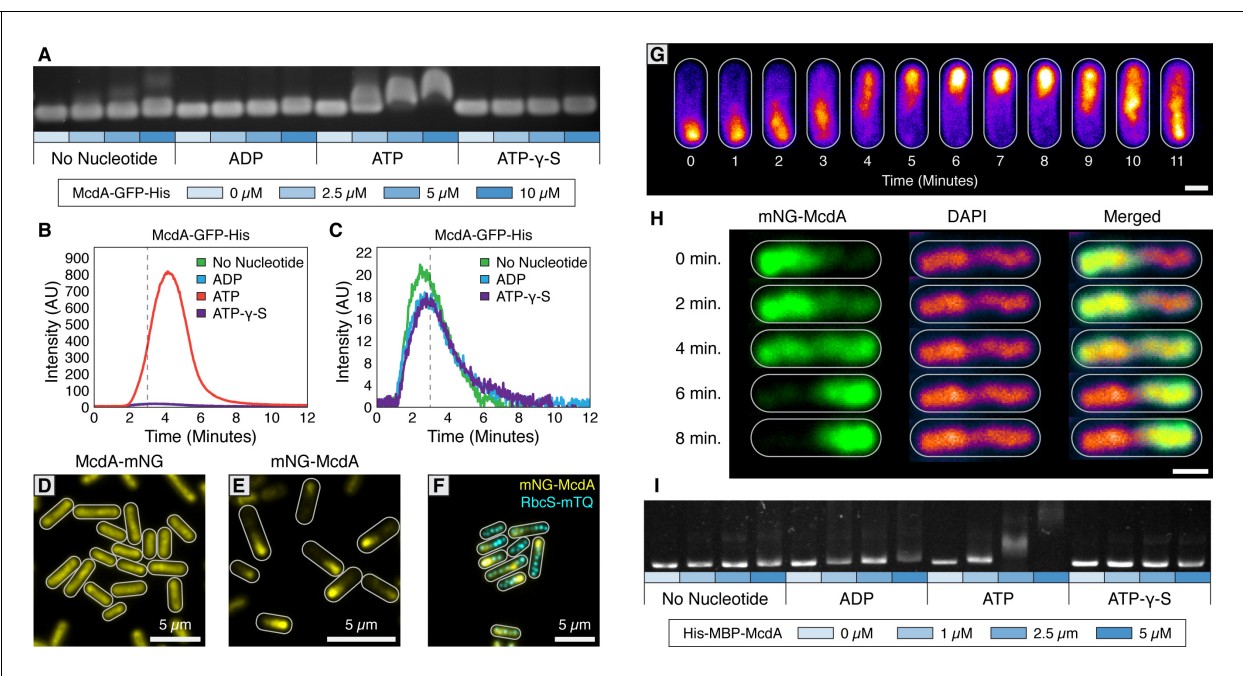

**Figure 1.** The ParA-like protein, McdA, binds nonspecifically to nucleoid DNA in the presence of ATP. (**A**) Purified McdA-GFP-His slows the migration of DNA in an ATP-dependent manner in an electrophoretic mobility shift assay (EMSA). (**B**) In flow cell experiments containing a carpet of non-specific DNA, McdA-GFP-His is preferentially retained by binding the DNA-carpet only in the presence of ATP, and releases when switching to a wash buffer after 3 min. (**C**) Magnified version of Panel B to show no protein binding without ATP. See *Video 1*. (**D**) Native McdA-mNG does not oscillate, while the N-terminally tagged variant (**E**) native mNG-McdA displays robust oscillations in *S. elongatus*. (**F**) mNG-McdA (yellow) oscillates to position carboxysomes (RbcS-mTQ - cyan). (**G**) mNG-McdA waves concentrate predominantly along the central axis of the cell, and do not concentrate at the periphery near the cell membrane. Scale bar = 1 μm. (**H**) Oscillation of mNG-McdA colocalizes with DAPI staining of nucleoid DNA, which is also centrally localized and is excluded from peripheral thylakoid membranes. Scale bar = 1 μm. (**I**) Purified His-MBP-McdA slows the migration of DNA in an ATP-dependent manner in an EMSA.

DOI: https://doi.org/10.7554/eLife.39723.003

The following source data and figure supplements are available for figure 1:

**Figure supplement 1.** mNG-McdA oscillates and mNG-McdB is recruited to carboxysomes.

DOI: https://doi.org/10.7554/eLife.39723.004

**Figure supplement 1—source data 1.** Source data for panel C.

DOI: https://doi.org/10.7554/eLife.39723.005

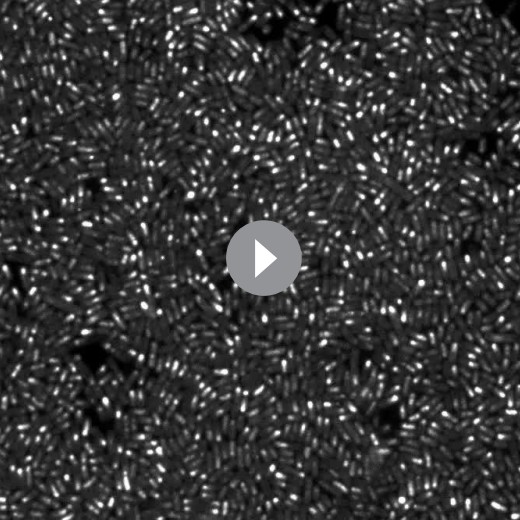

**Video 1.** McdA binds DNA only in the presence of ATP. After 3 min, a wash buffer is flowed in.
DOI: https://doi.org/10.7554/eLife.39723.006

previously reported using McdA-GFP [*Savage et al., 2010*]) that formed a bimodal distribution of signal intensity (*Figure 1EG*, *Figure 1—figure supplement 1B* and *Video 2*).

A carboxysome reporter was then generated by insertion of an additional copy of the small subunit of RuBisCO (RbcS) fused at the C-terminus to mTurquoise2 (mTQ) and expressed using a second copy of the native *rbcS* promoter. In this line, an average of 2 carboxysomes per micron of cell length was observed (*Figure 1—figure supplement 1C*). Cells bearing the mNG-McdA construct maintained carboxysome positioning along the longitudinal axis (≥99% of cells; n = 374 cells) (*Figure 1F*, *Figure 1—figure supplement 1D* and *Video 3*), indicating the N-terminal fusion fully complemented McdA's known functions. To assay *in vivo* whether mNG-McdA could be binding the nucleoid, we stained the cyanobacterial nucleoid with 4′,6-Diamidine-2′-phenylindole dihydrochloride (DAPI) and recorded the mNG-McdA signal as it traversed the length of the cell. We found that the topology of the mNG-McdA signal closely resembled that of the DAPI-stained nucleoid (*Figure 1H*), providing additional evidence that the nucleoid is the surface upon which McdA dynamics are occurring. Since an N-terminal fusion of McdA was more functional *in vivo*, we performed an EMSA with an N-terminal fusion of McdA. While we found that wild-type McdA and His-mNG-McdA were insoluble and prone to degradation, an N-terminal fusion of McdA to Maltose Binding Protein (MBP) (His-MBP-McdA) was highly soluble. Consistent

**Video 2.** Oscillation of mNG-McdA is observed across a population of cells.
DOI: https://doi.org/10.7554/eLife.39723.007

with our McdA-GFP-His EMSA, His-MBP-McdA shifted nsDNA in an ATP-dependent manner (*Figure 1I*). Together, our results demonstrate that ATP-bound N- or C-terminally tagged McdA binds DNA and McdA-GFP-His does not display indications of polymer formation at the resolution limits of our microscope.

## A previously uncharacterized protein is essential for emergent McdA dynamics

Traditional ParA-family members require cognate ParB proteins to stimulate their ATPase activity and promote oscillatory dynamics. Yet no ParB-like ortholog has been identified for McdA. Although no obvious chromosomally-encoded homolog of *parB* could be detected in *S. elongatus*, we identified a *parB*-like gene (Synpcc7942_B2626) on the large plasmid (pANL) (*Figure 2A*). However, deletion of pANL *parB* did not disrupt oscillation of mNG-McdA (≥99% of cells; n = 554 cells) (*Figure 2B* and *Figure 1—*

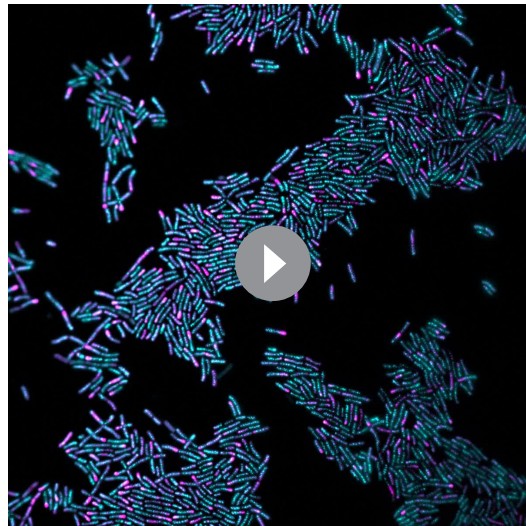

**Video 3.** Oscillation of mNG-McdA (magenta) occurs while carboxysomes (blue) are fluorescently labeled.
DOI: https://doi.org/10.7554/eLife.39723.008

*figure supplement 1E*). Two additional hypothetical genes were then selected due to their proximity to the *mcdA* gene, Synpcc7942_1834 and Synpcc7942_1835 (*Figure 2A*). While deletion of *Synpcc7942_1835* had no observable effect on mNG-McdA oscillation ($\geq$99% of cells; n = 834 cells) (*Figure 2C* and *Figure 1—figure supplement 1F*), deletion of *Synpcc7942_1834* resulted in complete loss of mNG-McdA dynamics ($\geq$99% of cells; n = 373 cells). In the $\Delta$*Synpcc7942_1834* background, mNG-McdA concentrated in the center of the cell in the vicinity of the nucleoid, but without any consistent asymmetrical patterning (*Figure 2D* and *Figure 1—figure supplement 1G*). To more descriptively designate the activities we observe for *Synpcc7942_1834* in this work, we will hereafter refer to this gene as <u>m</u>aintenance of <u>c</u>arboxysome <u>d</u>istribution <u>B</u> (*mcdB*).

Bioinformatic analysis of the McdB protein by BlastP (protein-protein blast) revealed that McdB lacked homology to any known ParB family member, nor any identifiable conserved regions with known ParB proteins. We therefore used the jPred4 platform (*Drozdetskiy et al., 2015*) to analyze this 17 kDa protein, which predicted that McdB possesses a secondary structure consisting mainly of alpha-helices, a highly charged N-terminal $a_1$-helix, and a coiled-coil C-terminal $a_8$-helix (*Figure 2E*). Because McdB appears to be a novel protein and there are no characterized proteins with comparable sequence, Phyre2 (*Kelley et al., 2015*) was unable to generate a reliable protein homology model.

We sought to determine if McdA and McdB directly interact by performing a bacterial two-hybrid assay (B2H) between N- and C-terminally tagged McdA and McdB (*Figure 2F*). McdA and McdB were able to self-associate in the B2H analysis. Self-association of C-terminally tagged McdA proteins was faint, but confirmed on X-gal plates (*Figure 2—figure supplement 1A*). We also observed a reciprocal interaction between N-terminally tagged McdA and N-terminally tagged McdB (*Figure 2F*). However, C-terminally tagged McdB failed to show an interaction with McdA, while C-terminally tagged McdA association with McdB was dependent upon expression conditions. These results suggest that N-terminally tagged McdA only interacts with N-terminally tagged McdB, while C-terminal fusions of either protein partially disrupts function, consistent with our *in vivo* observation of mNG-McdA dynamics in comparison to McdA-mNG (*Figure 1DE*).

The ParA/MinD family of ATPases are defined by the presence of two lysine residues within their deviant Walker-A motif (KGGXXGKT) required for dimerization, ATP-binding, and ATP hydrolysis (*Lutkenhaus, 2012*). Interestingly, *S. elongatus* McdA lacks the signature amino terminal lysine residue (*Figure 2G*), suggesting that McdA might have an activity uncharacteristic of proteins from this family. Therefore, we set out to determine if McdA displayed ATPase activity. His-MBP-McdA displayed strong ATPase activity alone, significantly higher (>200 fold) than that of traditional ParA family ATPases SopA of F plasmid and ParA of P1 plasmid (*Figure 2G* and *Figure 2—figure supplement 1B*). Because the ATPase activity was uncharacteristically high, we confirmed that the measured ATPase activity co-eluted with His-MBP-McdA from a size exclusion chromatography column and could not be attributed to a contaminating protein (*Figure 2—figure supplement 1C*). Relative to the constant specific activity of F SopA-His and P1 ParA with increasing protein concentrations (*Figure 2—figure supplement 1D*), the specific ATPase activity of His-MBP-McdA declined at higher protein concentrations (*Figure 2—figure supplement 1E*). This decrease in ATPase activity was not due to substrate limitation during the course of the *in vitro* assay, as ATP was provided in excess, but could be indicative of a regulatory mechanism or product inhibition that is not characteristic of traditional ParA family members.

ParB partners stimulate the ATPase activity of their cognate ParA synergistically with nsDNA (*Davis et al., 1992*; *Bouet and Funnell, 1999*; *Bouet et al., 2000*). ParB stimulation is suggested to be coupled to ParA depletion on the nucleoid in the vicinity of ParB-bound cargo, as ADP does not support ParA binding to nsDNA (*Leonard et al., 2005*; *Hester and Lutkenhaus, 2007*; *Castaing et al., 2008*; *Vecchiarelli et al., 2010*). We tested whether McdB-His could stimulate McdA ATPase activity, analogously to traditional ParB members. When adding only nsDNA or McdB-His to the reactions, we observed very mild stimulation of His-MBP-McdA ATPase activity (*Figure 2H*). When both nsDNA and McdB-His were added simultaneously, the ATPase activity of His-MBP-McdA was further stimulated (*Figure 2H* and *Figure 2—figure supplement 1G*). An alternative preparation of McdA which was tagged at the C-terminus with GFP also exhibited intrinsically high ATPase activity that could be stimulated with the addition of nsDNA (*Figure 2—figure supplement 1F*), but addition of McdB-His did not further increase ATPase activity, in agreement with our

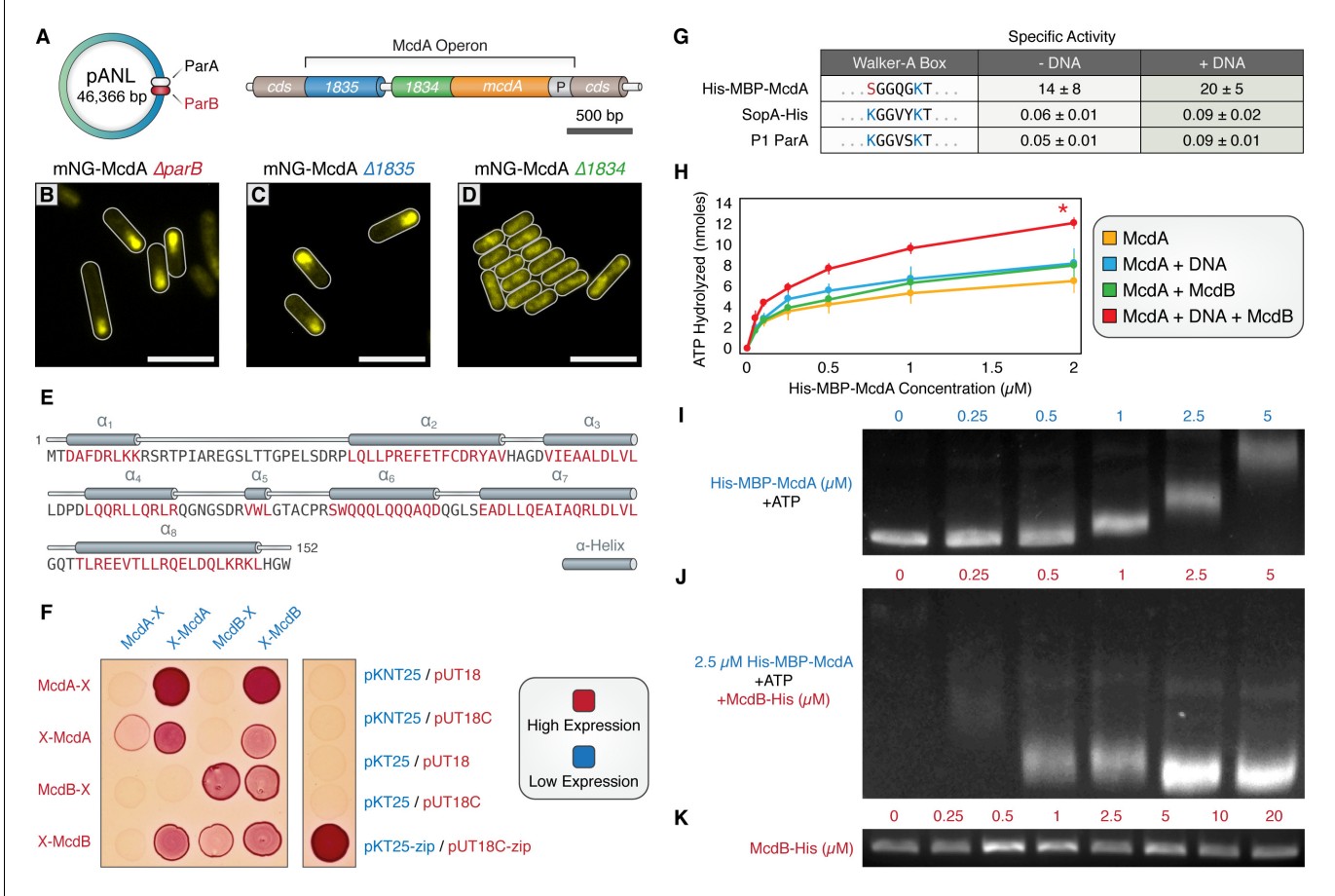

**Figure 2.** A small, carboxysome-localized protein is required for McdA oscillatory dynamics. (**A**) Candidate genes controlling McdA dynamics on either the native plasmid (left) or within the genomic neighborhood of McdA. (**B**) Deletion of pANL *parB* or (**C**) *Synpcc7942_1835* has no effect on mNG-McdA dynamics. (**D**) Deletion of *Synpcc7942_1834* collapses mNG-McdA oscillatory dynamics. All scale bars = 5 µm. (**E**) *Synpcc7942_1834* secondary structure prediction: residues in red are predicted to form α-helical secondary structure. (**F**) Bacterial two-hybrid between McdA and McdB tagged at their N-termini (X-McdA, X-McdB) or C-termini (McdA-X, McdB-X) on a MacConkey agar plate. Image representative of 3 independent trials. (**G**) The deviant Walker A-box motif and specific activities of McdA, F SopA-His and P1 ParA with and without DNA. (**H**) ATPase activity assays of His-MBP-McdA reveals stimulatory roles of DNA and McdB-His. (**I**) His-MBP-McdA slows the migration of DNA in an ATP-dependent manner in an EMSA. (**J**) Increasing concentrations of McdB-His resolves the gel shift provided by 2.5 µM of His-MBP-McdA and ATP. (**K**) Titration of McdB-His shows it does not directly bind nsDNA.

DOI: https://doi.org/10.7554/eLife.39723.009

The following source data and figure supplements are available for figure 2:

**Figure supplement 1.** McdA Activity.
DOI: https://doi.org/10.7554/eLife.39723.010

**Figure supplement 1—source data 1.** Source data for panel B.
DOI: https://doi.org/10.7554/eLife.39723.011

**Figure supplement 1—source data 2.** Source data for panel D.
DOI: https://doi.org/10.7554/eLife.39723.012

**Figure supplement 1—source data 3.** Source data for panel F.
DOI: https://doi.org/10.7554/eLife.39723.013

**Figure supplement 1—source data 4.** Source data for panel G.
DOI: https://doi.org/10.7554/eLife.39723.014

prior data (*Figures 1D* and *2F*, *Figure 1—figure supplement 1A*) that suggests C-terminal fusions of McdA may prevent interactions with McdB.

In comparison to classic examples within the ParA/ParB family, McdB's stimulation of McdA ATPase activity is relatively mild (2–3 fold; *Figure 2G*). We therefore assayed if this stimulatory effect

was sufficient to influence McdA's binding to DNA substrates using a gel shift assay. As shown previously (*Figure 1I*), with ATP present, non-specific DNA fragments exhibit slowed mobility in the presence of increasing concentrations of purified His-MBP-McdA (*Figure 2I*). Conversely, when the experiment was conducted with a constant concentration of His-MBP-McdA (2.5 µM), the shift in DNA mobility was reversed by addition of increasing amounts of McdB-His (*Figure 2J*). This demonstrates that McdB favors the release of McdA from DNA. McdB-His alone did not exhibit any DNA binding activity in our gel shift assay (*Figure 2K*). Taken together, these results further support a direct interaction between McdA and McdB, and suggest that McdB drives the removal of McdA from its DNA substrate; contributing to the emergent oscillatory dynamics of McdA we observe *in vivo*.

## McdB localizes to carboxysomes via interaction with major and minor shell components

We next sought to elucidate McdB's localization *in vivo*. Similar to our native McdA reporters, we generated N- and C-terminal mNG fusions of McdB in its native genomic locus downstream of *mcdA*. C-terminal fusions of McdB displayed a diffuse localization with random punctate-like patterns (≥99% of cells; n = 371 cells) (*Figure 3A* and *Figure 3—figure supplement 1A*). In contrast, N-terminal mNG-McdB was observed as multiple discrete fluorescent foci near the central longitudinal axis of the cell (≥99% of cells; n = 699 cells), a result that strongly resembles the localization pattern of native carboxysomes (*Figure 3B* and *Figure 3—figure supplement 1B*). We confirmed co-localization of McdB with carboxysomes by co-expression of the carboxysome fluorescent reporter in the mNG-McdB strain. Both the mNG-McdB and RbcS-mTQ signals strongly colocalized (≥99% of cells; Pearson's Correlation Coefficient [PCC]=0.92; n = 316 cells) as fluorescent foci near the long central axis of the cell (*Figure 3C* and *Figure 3—figure supplement 1C*).

Next, we investigated if McdB's interaction with the carboxysome is direct, and if so, what carboxysome components bind McdB. During biogenesis, carboxysomes first form a core structure containing RuBisCO and carbonic anhydrase, which are coordinated into an ordered array through interactions with CcmM (*Figure 3D*) (*Long et al., 2007*; *Cot et al., 2008*; *Long et al., 2010*; *Cameron et al., 2013*). This core is thought to recruit outer shell proteins through the mediating protein CcmN, thereby forming the carboxysome coat (*Fan et al., 2012*; *Kinney et al., 2012*) (*Figure 3D*). CcmK2 is the dominant shell protein that composes the facets of the shell, and which has been shown to directly interact with CcmN (*Kinney et al., 2012*). Along with CcmK2, proteins CcmO, CcmL, CcmK3, CcmK4, and CcmP are also recruited to complete compartmentalization (*Figure 3E*) (*Tanaka et al., 2008*; *Tanaka et al., 2009*; *Rae et al., 2012*; *Cai et al., 2013*), although the relative arrangement of these structural components of the shell remains uncertain. We explored if the outer shell proteins of the carboxysome could be involved in recruiting McdB through a bacterial two-hybrid screen. Using N- or C-terminally tagged McdA or McdB as bait, the assay suggested only N-terminally tagged McdB interacts with the shell proteins CcmK2, CcmK3, CcmK4, CcmL, and CcmO, but not CcmP (*Figure 3F* and *Figure 3—figure supplement 1D*). In contrast, we did not find evidence for direct interaction between McdA and carboxysome shell proteins (*Figure 3F* and *Figure 3—figure supplement 1D*).

## Carboxysome formation is required for the emergent oscillations of McdA

To further investigate the association of McdB with the carboxysome, we examined mNG-McdB dynamics in a *S. elongatus* background that lacks functional carboxysomes. While carboxysomes are essential for growth under an ambient atmosphere, mutants deleted for the Carbon Concentrating Mechanism (*ccm*) operon (Δ*ccmK2LMNO*; *Figure 3G*) can be recovered in high $CO_2$ (*Price et al., 1993*; *Cameron et al., 2013*). We therefore examined the localization of mNG-McdB in a Δ*ccmK2LMNO* background and found that mNG-McdB signal was diffuse (≥99% of cells; n = 389 cells) in the absence of carboxysomes (*Figure 3H* and *Figure 3—figure supplement 1E*), further indicating a recruitment of McdB to assembled carboxysomes. As expected, RbcS-mTQ signal was also diffuse in Δ*ccmK2LMNO* cells, confirming the absence of carboxysomes (*Figure 3H* and *Figure 3—figure supplement 1E*). Interestingly, in the absence of carboxysomes, mNG-McdA did not oscillate and formed a homogenous distribution along the nucleoid similar to that of our mNG-

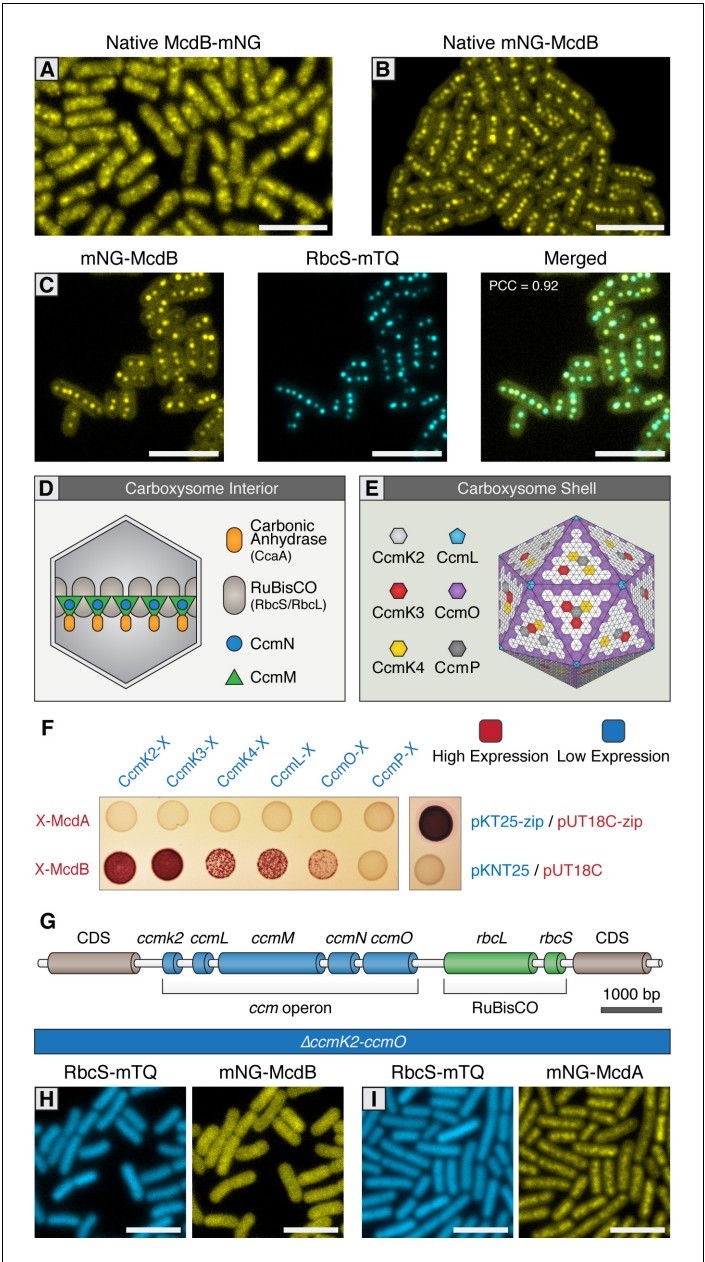

**Figure 3.** McdB interacts with the carboxysome shell. (**A**) Native McdB-mNG forms diffuse and punctate-like structures, while (**B**) native mNG-McdB localizes to centralized foci in *S. elongatus* cells. (**C**) mNG-McdB and a carboxysome reporter - RbcS-mTQ - colocalize. Pearson correlation coefficient (PCC) is 0.92. (**D**) Cartoon depiction of the internal and (**E**) external shell components of the cyanobacterial carboxysome, adapted from models presented in *Rae et al., 2013*. Note that the relative geometric arrangement of some shell components to one another remain somewhat speculative. (**F**) Bacterial two-hybrid of McdA and McdB against carboxysome shell proteins. Image representative of 3 independent trials. (**G**) Cartoon schematic of the *S. elongatus* carboxysome operon. (**H**) In a ΔccmK2-O mutant without carboxysomes, mNG-McdB is not localized to central puncta but is diffuse, while (**I**) mNG-McdA oscillatory dynamics are lost. All scale bars = 5 μm.

DOI: https://doi.org/10.7554/eLife.39723.015

The following figure supplement is available for figure 3:

**Figure supplement 1.** McdA/B bacterial two-hybrid against carboxysome shell proteins.

DOI: https://doi.org/10.7554/eLife.39723.016

McdAΔ*mcdB* strain (≥99% of cells; n = 227 cells) (*Figure 3I*, *Figure 3—figure supplement 1F* and *Figure 2D*). This result suggests that the interaction and/or concentration of McdB onto carboxysomes is required for self-organization of McdA and could be an important prerequisite for strong stimulation of McdA ATPase activity.

## Carboxysome positioning is disrupted in McdA/McdB Mutants

Since we found that both McdA and McdB are implicated in the regulation of carboxysome positioning, we next investigated how carboxysomes were distributed in strains lacking these proteins. To reduce potential off-target effects in the Δ*mcdA* and Δ*mcdB* lines, we generated knockouts in a manner designed to minimize alterations in expression of neighboring genes. This included insertion of the kanamycin resistance cassette outside of the McdA operon, and duplication of the *mcdA* promoter upstream of the *mcdB* gene in constructs where interruption of *mcdA* might be expected to disrupt downstream gene expression (see *Figure 4AB*). Simultaneously, we also inserted RbcS fused to mOrange2 (mO) expressed from the native *rbcS* promoter to visualize carboxysomes (*Figure 4ABC*).

As before, mNG-McdA distributed along the nucleoid and did not oscillate in Δ*mcdB* lines (≥99% of cells; n = 373 cells) (*Figure 4A*), and carboxysomes were observed as large irregularly shaped polar fluorescent foci with smaller randomly distributed signals within the cell (*Figure 4A*). Consistent with our proposed role for McdB in removing McdA from the nucleoid, we did not observe depleted McdA signal on the nucleoid in the vicinity of carboxysomes within a Δ*mcdB* background. In Δ*mcdA* lines, mTQ-McdB still localized to carboxysomes, indicating McdA is not required for the association (≥99% of cells; PCC = 0.85; 416 cells). Carboxysomes also formed large fluorescent foci in Δ*mcdA* lines and, where multiple foci could be resolved, they frequently clustered in close proximity rather than distributing throughout the cell (*Figure 4B*), similarly to Δ*mcdB* lines. In the absence of both *mcdA* and *mcdB*, carboxysomes appeared as irregular foci of varying sizes randomly distributed through the cell (≥99% of cells; n = 399 cells) (*Figure 4C*). The fluorescence intensity from the RbcS-mO reporter was also unexpectedly ~4 fold weaker in these lines, likely due to unintended read-through effects from the mcdA promoter.

We next investigated carboxysome positioning in McdA and McdB overexpression lines. For these experiments, we inserted the RbcS-mO fluorescent reporter in neutral site 1, a genomically neutral locus (*Clerico et al., 2007*), and overexpressed either mNG-McdA or mTQ-McdB from a synthetic riboswitch (*Nakahira et al., 2013*) inserted into neutral site 2 (*Clerico et al., 2007*) (*Figure 4DE*). Upon the overexpression of mNG-McdA, we observed loss of McdA oscillation and the formation of large irregularly-shaped RbcS-mO fluorescent foci reminiscent of Δ*mcdA*, Δ*mcdB* and Δ*mcdAB* lines (≥99% of cells; n = 340 cells) (*Figure 4D*). With endogenous levels of McdB present, the McdA signal was generally depleted in the vicinity of these carboxysome aggregates. Upon overexpression of mTQ-McdB, we observed mTQ-McdB signal colocalized at irregularly-shaped RbcS-mO fluorescent foci as well as diffuse within the cell (≥99% of cells; PCC = 0.80; n = 345 cells) (*Figure 4E*). Moreover, we occasionally observed (<10% of cells) much larger RbcS-mO bar-shaped structures within the cell (*Figure 4E*; yellow arrow), indicating that McdB levels might influence carboxysome size and ultrastructure.

## Carboxysome ultrastructure is disrupted in McdA/McdB Mutants

Carboxysome distribution displayed some variation even in wildtype strains, therefore we quantified carboxysome distributions in hundreds of cells utilizing MicrobeJ to automatically detect and characterize fluorescent carboxysome foci (*Ducret et al., 2016*). In our RbcS-mO only reporter line, carboxysomes were observed predominately along the central axis and mean focus diameter was 140 nm ± 100 nm (*Figure 5A,G*). In comparison to this strain, carboxysomes in Δ*mcdA* (510 nm ± 270 nm), Δ*mcdB* (350 nm ± 190 nm), and Δ*mcdAB* (370 nm ± 250 nm) lines had a much broader distribution off the central axis with larger mean foci diameters and deviations (*Figure 5BCDG*). Likewise, overproduction of McdA produced carboxysome distributions off the central axis and a mean foci diameter (320 nm ± 300 nm) similar to those of deletion lines (*Figure 5BCDEG*). Lastly, McdB overexpression produced carboxysomes that were distributed mostly along the central longitudinal axis, but exhibited the largest mean foci diameter and deviation (530 nm ± 470 nm) (*Figure 5FG*).

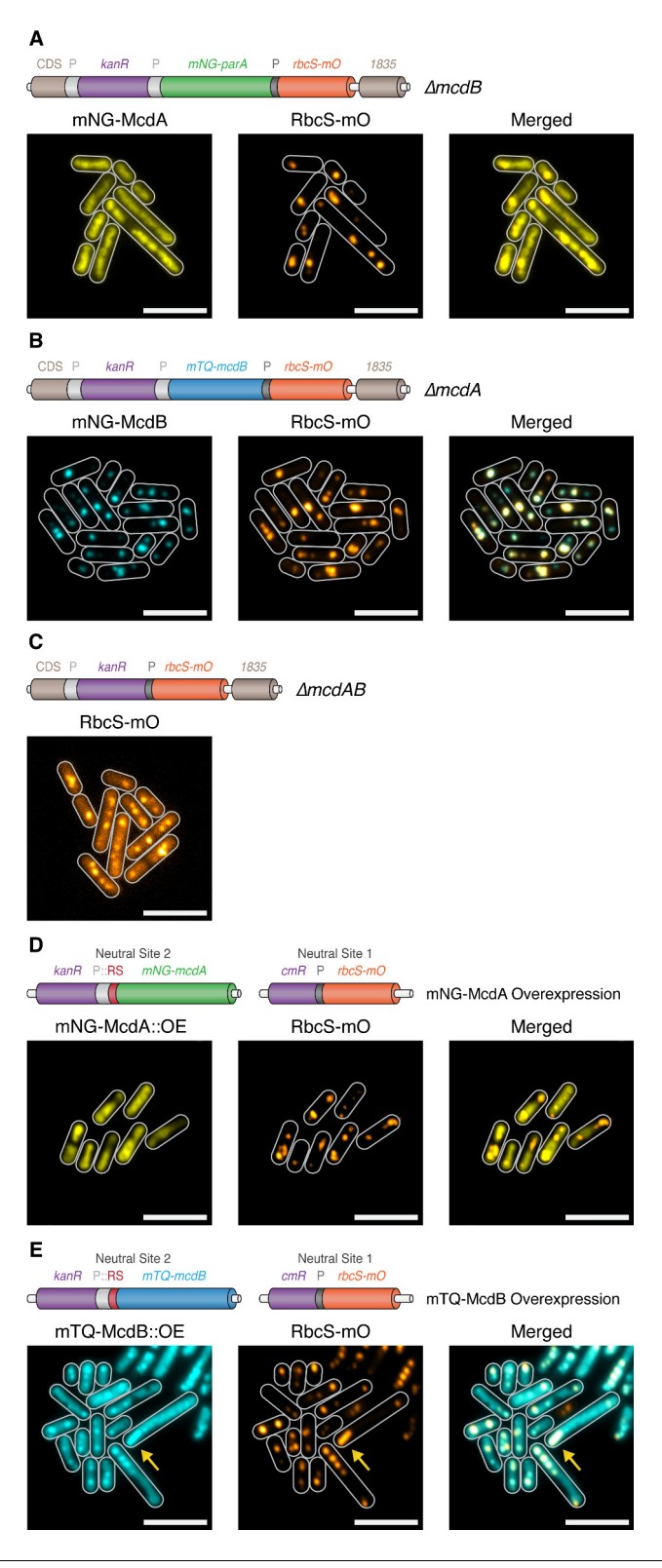

**Figure 4.** McdA and McdB are essential for distributing carboxysomes. (**A**) Carboxysome (orange; rbcS-mO) and mNG-McdA (yellow) distribution in a Δ*mcdB* background. Cartoon schematics (top) of the genetic construct are depicted, mcdA promoter 'P'=light grey, rbcS promoter = dark grey. mNG-McdA signal did not oscillate and was found distributed on the nucleoid (**B**) Carboxysome and mTQ-McdB (cyan) distribution in a Δ*mcdA* background. *Figure 4 continued on next page*

*Figure 4 continued*

mTQ-McdB colocalizes to *S. elongatus* carboxysomes (RbcS-mO) (PCC = 0.85). (**C**) Carboxysome distribution with *mcdAB* deleted. (**D**) Carboxysome positioning is disrupted and oscillation of mNG-McdA does not occur upon mNG-McdA overexpression. mNG-McdA signal did not oscillate and was found distributed on the nucleoid. Some carboxysomes had mNG-McdA signal depleted in their vicinity. Riboswitch (RS) = red. (**E**) Bar carboxysome structures (yellow arrow) and loss of carboxysome positioning are observed upon overexpression of mTQ-mcdB. mTQ-McdB signal colocalized with RbcS-mO signal (PCC = 0.80). All scale bars = 5 µm.

DOI: https://doi.org/10.7554/eLife.39723.017

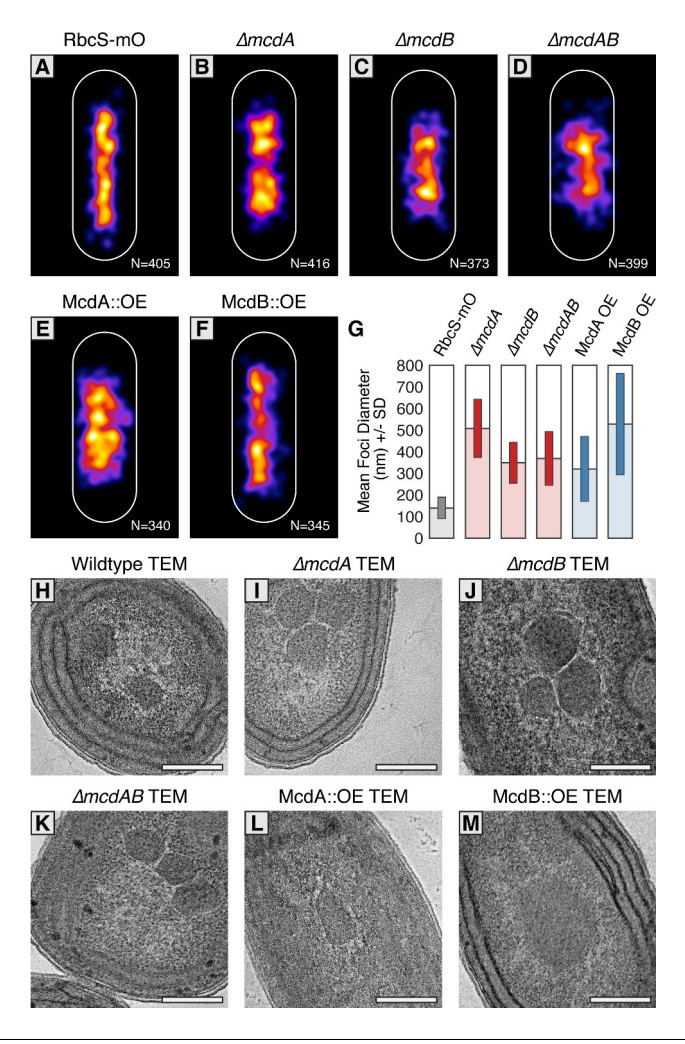

**Figure 5.** McdAB physically separate and regulate carboxysome ultrastructure. (**A–F**) MicrobeJ average distribution heat-map of carboxysomes in the indicated genetic background strains. Quantity of cells measured in lower right corner. (**G**) Mean RbcS-mO foci area and standard deviation for wildtype and the indicated mutant strains. (**H–M**) Electron micrographs of carboxysomes from wildtype and mutant strains. All scale bars = 250 nm.

DOI: https://doi.org/10.7554/eLife.39723.018

The following figure supplement is available for figure 5:

**Figure supplement 1.** Quantification of carboxysome size from TEM.

DOI: https://doi.org/10.7554/eLife.39723.019

This data suggests that in addition to spatially regulating carboxysome distributions, McdA and McdB could influence carboxysome size and/or ultrastructure.

To differentiate whether the changes in size of RbcS-mO foci were due to clustering of multiple carboxysomes or changes in carboxysome ultrastructure, we used Transmission Electron Microscopy (TEM). In contrast to faithfully distributed carboxysomes in our wildtype TEM images (*Figure 5H* and *Figure 5—figure supplement 1B*), we observed multiple carboxysomes tightly clustered in our Δ*mcdA*, Δ*mcdB*, and Δ*mcdAB* strains (*Figure 5IJK* and *Figure 5—figure supplement 1ACDE*). This observation suggested McdA and McdB are required to separate neighboring carboxysomes, or that newly synthesized carboxysomes may be incompletely detached from one another during biogenesis (*Cameron et al., 2013*; *Chen et al., 2013*) in the absence of McdA or McdB.

Upon overproduction of McdA, we observed irregularly-shaped carboxysomes with 'rounded' edges that tightly clustered (*Figure 5L* and *Figure 5-figure supplement 1AF*). It should be noted however, that this cell line displayed severe growth arrest and possessed an unusually abundant number of granules which we suggest could be polyphosphate bodies based on their structure and dense staining. Consistent with this observation, overproduction of ParA family proteins in many organisms is lethal (*Lasocki et al., 2007*). Most strikingly, carboxysomes did not cluster in McdB overproduction strains either, instead, carboxysome ultrastructure was dramatically altered. Unlike the classic icosahedral-shape, carboxysomes were observed as large irregular bar-like structures (*Figure 5M* and *Figure 5—figure supplement 1AG*). In some instances, these 'bar-carboxysomes' extended hundreds of nanometers (*Figure 4E*), resembling previous reports of improperly assembled carboxysomes in cells lacking *ccmL* (*Price and Badger, 1989*; *Cameron et al., 2013*). Together, these results highlight the importance of the McdAB system for spatially separating carboxysomes and regulating the underlying size and ultrastructure.

## Carboxysomes locally deplete McdA on the nucleoid and are required for McdA oscillation

In ParA-based plasmid partitioning, the low-copy number of plasmid cargo allows for the direct observation of how one or two plasmid copies influence the distribution of ParA on the nucleoid, and vice versa. Through *in vivo* and *in vitro* experimentation, it has been demonstrated that ParB-bound plasmids and chromosomes form ParA depletion zones by stimulating the release of ParA from the nucleoid in their vicinity (*Hatano et al., 2007*; *Ringgaard et al., 2009*; *Schofield et al., 2010*; *Hwang et al., 2013*; *Vecchiarelli et al., 2013*; *Vecchiarelli et al., 2014*). But in *S. elongatus*, carboxysome copy-number correlates with cell length and ranges between 3 to 15 copies (*Figure 1—figure supplement 1C*). How multiple carboxysomes are equally spaced along the cell-length by a global oscillation of McdA protein is not intuitively obvious. Therefore, we wished to determine if McdB-bound carboxysomes also cause similar local depletions of McdA on the nucleoid and whether oscillation of McdA was a requirement for carboxysome motion and equidistant positioning, even at low-copy numbers.

In time-lapse experiments of mNG-McdA and RbcS-mTQ strains, we frequently observed that carboxysomes move towards the highest local concentration of McdA on the nucleoid; this was especially apparent as the wavefront of the oscillating McdA pool approaches carboxysomes (*Videos 3–5*). The most rapid directed motions of carboxysomes were visible as the wavefront approaches a carboxysome, and then as the wavefront passes.

To explore this observation in more depth, we developed the ability to regulate the initiation of carboxysome formation as well as to modulate the number of carboxysomes per cell. To accomplish this, we replaced the native *ccmK2* promoter with a Ptrc promoter lacking the *lacI* repressor and attached a 5' synthetic riboswitch preceding *ccmK2* (RS::*ccmK2LMNO*; *Figure 6A*). In the absence of inducer, genes regulated by this riboswitch are tightly off, and expression is highly tunable with increasing concentrations of theophylline (*Nakahira et al., 2013*). In the absence of theophylline, we observed that RbcS-

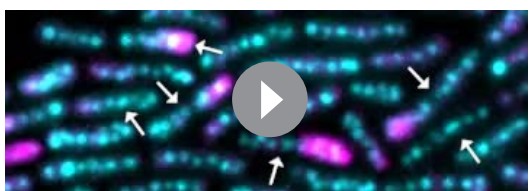

**Video 4.** Carboxysomes (blue; RbsS-mTQ) move towards increased concentrations of mNG-McdA (magenta); notable examples of this phenomenon are denoted by white arrows. Each frame is 30 seconds. DOI: https://doi.org/10.7554/eLife.39723.021

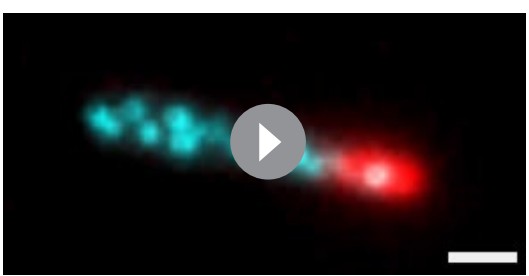

**Video 5.** Zoomed-in single cell representation of carboxysomes (blue; RbsS-mTQ) moving towards increased concentrations of mNG-McdA (red). Each frame is 15 seconds.
DOI: https://doi.org/10.7554/eLife.39723.022

mTQ signal was diffuse and mNG-McdA was distributed homogenously along the nucleoid (≥99% of cells; n = 204 cells) (*Figure 5—figure supplement 1H*), consistent with our prior results from ΔccmK2LMNO mutants (*Figure 3I*). When these strains were induced with either 400 µM or 600 µM theophylline, we were able to generate on average one or two carboxysomes per cell, respectively (*Figure 6BC*). In the presence of 1 carboxysome, mNG-McdA signal remained evenly distributed except for a depletion zone that correlated with the nucleoid region in the vicinity of the carboxysome (*Figure 6B*). Likewise, with two carboxysomes, mNG-McdA signal again distributed along the nucleoid but was depleted in areas correlating to carboxysomes (*Figure 6C*). In either case, mNG-McdA signal was highly reduced in areas of RbcS-mTQ signal (≥99% of cells; PCC = 0.20; n = 391 cells), indicating McdB-bound carboxysomes have mNG-McdA depleted in their vicinity. In this strain, we performed real-time imaging of mNG-McdA dynamics and RbcS-mTQ motion. In instances where cells contained two, closely spaced carboxysomes, one carboxysome could be clearly observed to move in the direction of the higher McdA concentration (*Figure 6D*). When a sufficient distance was reached between the two carboxysomes, mNG-McdA was re-recruited to the depleted nucleoid region between the two carboxysomes (*Figure 6D*). As mNG-McdA rebound the nucleoid, movement of the centralized carboxysome halted and slightly regressed back in the opposite direction (*Figure 6D*). This result is consistent with the Brownian-ratchet mechanism for genetic cargo movement towards the highest local concentration of ParA (*Vecchiarelli et al., 2010*; *Vecchiarelli et al., 2014*; *Hu et al., 2017*).

We next sought to determine if we could reconstitute carboxysome-dependent oscillation of McdA. Even at relatively high concentrations of theophylline inducer, our synthetic riboswitch was unable to generate wildtype quantities of carboxysomes. Therefore, we used a variant of a previously published approach (*Cameron et al., 2013*) by replacing the *ccmK2* promoter with the Ptrc promoter and inserted an upstream *lacI* repressor (*Figure 6E*). This promoter is generally capable of driving higher expression levels of gene targets. In the absence of Isopropyl β-D-1-thiogalactopyranoside (IPTG), some carboxysome formation was observed due to leaky expression of the Ptrc promoter in cyanobacteria (*Figure 6F*). Similar to prior results, mNG-McdA was depleted in the vicinity of carboxysomes (≥99% of cells; n = 453 cells) (*Figure 6F*). Following induction with IPTG, multiple carboxysomes formed throughout cells and mNG-McdA oscillations emerged (≥99% of cells; n = 439 cells; *Figure 6F*). Altogether, these experiments strongly indicate that McdB is concentrated upon carboxysomes and that this localized pool of McdB changes the dynamics of McdA bound to neighboring regions of the nucleoid. Under conditions where there are relatively few (1-3) carboxysomes, McdB appears to continuously stimulate the release of nearby McdA. It is only at higher numbers of carboxysomes (4+) when a self-organized oscillation of McdA from end-to-end of the nucleoid emerges. Moreover, we observe multiple instances of directed motion of carboxysomes towards increased McdA concentrations on the nucleoid, consistent with the Brownian-ratchet model of cargo movement.

## Carboxysomes are hexagonally arranged when crowded on a nucleoid

Carboxysomes are frequently described as being linearly arranged along the longitudinal axis of *S. elongatus*. In addition to linear distributions, we also routinely observed carboxysomes that were equidistant to each other, but no longer linearly arranged (*Figure 7A*). Instead, carboxysomes displayed a hexagonal packing phenomenon where the linear arrangement along the longitudinal axis looked kinked or displayed a zig-zag pattern. The hexagonal packing arrangement is non-intuitive assuming carboxysome distribution were to be based solely on either an underlying cytoskeleton or McdA oscillations, as these features are oriented longitudinally to the cell. Hexagonal packing was typically correlated with cells that had a high number of carboxysomes relative to the cell's length.

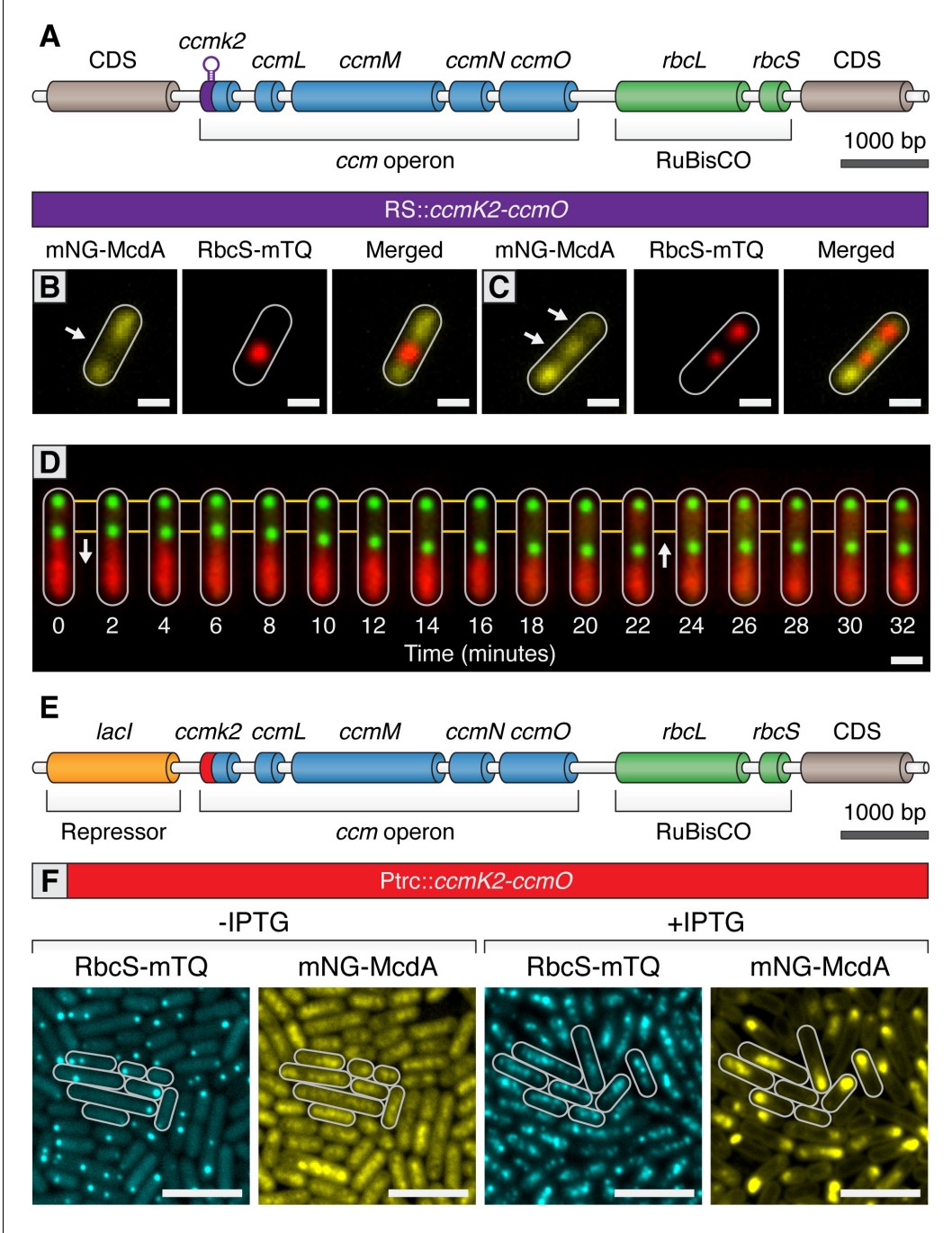

**Figure 6.** Carboxysomes locally deplete McdA from the nucleoid and cause McdA oscillation at high-copy number. (**A**) Cartoon schematic of the inducible carboxysome operon with the synthetic, theophylline-inducible riboswitch proceeding *ccmK2*. Induction of carboxysome biogenesis with either (**B**) 400 μM or (**C**) 600 μM theophylline leads to the formation of one or two carboxysomes (red; RbcS-mTurq). mNG-McdA (yellow) is depleted from the vicinity of nearby carboxysomes (white arrows). Scale bar = 1 μm. (**D**) In a representative timelapse imaging of a newly formed carboxysome (green), the central carboxysome moves towards (white arrow) a cellular location with increased mNG-McdA (red). When sufficient distance is obtained between carboxysomes (~20 min) mNG-McdA begins to be recruited between them. Scale bar = 1 μm. (**E**) Cartoon schematic of carboxysome operon with Ptrc promoter and *lacI* proceeding *ccmK2*. (**F**) mNG-McdA oscillations are reconstituted following addition of IPTG and the assembly of multiple carboxysomes. Scale bar = 5 μm.
DOI: https://doi.org/10.7554/eLife.39723.020

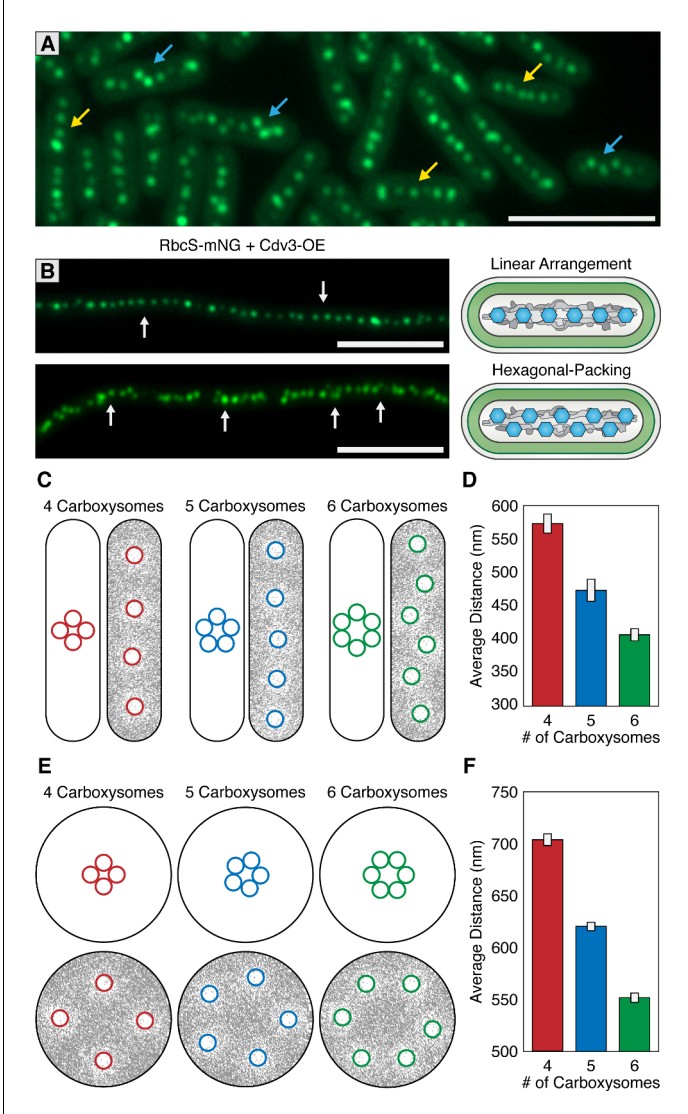

**Figure 7.** McdAB accounts for linear and hexagonal packing of carboxysomes. (**A**) In a field of *S. elongatus* cells, carboxysomes are found in either a linear (yellow arrow) or hexagonal (blue arrow) arrangement. Scale bar = 5 µm. (**B**) Linear or hexagonal arrangement of carboxysomes in filamentous cells (white arrows). Scale bar = 5 µm. (**C**) Reaction-diffusion simulations of 4, 5, or six carboxysomes on a rounded-rectangle surface. Positions of carboxysomes in representative simulations indicated at the start of the simulation (left) and after the simulation has reached steady state (right). Grey = McdA. White = Nucleoid. (**D**) Average distance between carboxysomes. (**E**) Reaction-diffusion simulations of 4, 5, or six carboxysomes upon a round surface. Grey = McdA. White = Nucleoid. (**F**) Average distance between carboxysomes. Error bars = standard deviation.
DOI: https://doi.org/10.7554/eLife.39723.023

The following figure supplement is available for figure 7:

**Figure supplement 1.** Carboxysomes fall between nucleoids in elongated cells lacking McdA or McdB.
DOI: https://doi.org/10.7554/eLife.39723.024

However, it is difficult to ascertain if this different packing arrangement is due solely to carboxysome number, or any number of other factors that could be differentially regulated between distinct cells.

To better understand this hexagonal packing phenomenon, we examined carboxysome positioning in hyperelongated cells because these cells have unusual nucleoid features useful to supplement our observations in cells of wildtype length (3–6 µm). Importantly, such hyperelongated *S. elongatus* cells are known to contain nucleoid clusters which are separated by intermittent cytoplasmic gaps

that physically separate the nucleoid clusters from one another (*Miyagishima et al., 2005*); *Figure 7—figure supplement 1*). One targeted method to increase cell length is to overexpress the FtsZ regulatory protein Cdv3, which we have previously reported to cause division arrest and subsequent cell elongation up to 2 mm (*Jordan et al., 2017*; *MacCready et al., 2017*). In cells elongated by this method, we observed both linear and hexagonal carboxysome packing (*Figure 7B*), sometimes observing different packing arrangements on neighboring nucleoid clusters within the same cell (*Figure 7—figure supplement 1D*). Carboxysomes in hyperelongated cells always co-localized with a nucleoid cluster, as visualized by DAPI staining (*Figure 7—figure supplement 1A*), and were never observed in the gaps between clusters, consistent with a role for McdAB in tethering carboxysomes to DNA. By contrast, in a ΔmcdA or ΔmcdB background, when we induced hyperelongation by expressing Cdv3, carboxysomes were frequently observed in these cytoplasmic gap regions (*Figure 7-figure supplement 1BC*).

In individual hyperelongated cells, we frequently observed carboxysomes both in linear and hexagonal-packing arrangements within the same cell but on different nucleoid clusters (*Figure 7—figure supplement 1D*). Because cells containing both linear and hexagonal packing arrangements share the same cytosol, it is unlikely that the carboxysome packing is regulated by a global change within a cell (such as a diffusible factor). Instead, we found once again that the hexagonal packing was typically observed when the number of carboxysomes were higher on a given nucleoid cluster. These results suggest that carboxysome packing arrangement may be a self-emergent property related to the density of carboxysomes on a given nucleoid surface area.

## The Brownian-ratchet model is sufficient to explain carboxysome distributions

To assess if the Brownian-ratchet model of carboxysome positioning could account for both carboxysome spacing and patterning (i.e. linear vs. hexagonal), we turned to an established *in silico* mathematical model that has successfully described several aspects of the Brownian-ratchet mechanism for ParA-mediated partitioning of plasmids (*Hu et al., 2017*). Since we have yet to determine which biochemical parameters of the Mcd system differ from that of traditional Par systems, in this treatment, we simply increased the number of cargo copies on the nucleoid matrix while keeping all other biochemical parameters as previously described so as to determine if increasing cargo copy number is enough to convert linear positioning into hexagonal packing. We programmed the geometry of the nucleoid surface area (2.5 µm by 0.6 µm rounded rectangle) and carboxysome cargo (175 nm) based off of previously measured values of wildtype *S. elongatus* (*Rae et al., 2012*; *Murata et al., 2016*). All simulations are initiated with tightly clustered carboxysomes near the center of the nucleoid (*Figure 7C*; left images), but carboxysomes are allowed to travel towards the highest gradient of McdA using the previously-established parameter values.

With the Brownian-ratchet model, five or less carboxysomes will linearly distribute on a rectangular surface representative of *S. elongatus'* nucleoid (*Figure 7C*). As cargo number increases the linear arrangement is maintained, but with tighter spacing (*Figure 7CD*). However, above a certain density threshold (six or more cargos on the same nucleoid under our simulation parameters), cargo positioning switches from a linear arrangement to hexagonal packing, reminiscent of the *in vivo* distributions (*Figure 7ABCD*). This change in packing arrangement can be understood if each carboxysome is independently seeking the highest local concentration of McdA on the nucleoid: as carboxysome density increases, a staggered conformation maintains the maximal nearest-neighbor distance. As many other cyanobacterial species exhibit spherical morphology, including the model *Synechocystis sp. PCC 6803*, we also examined the predicted distribution of carboxysomes upon a 1.7 µm circular nucleoid (*Figure 7EF*). We suggest that the linear arrangement of carboxysomes in rod-shaped cells is largely a byproduct of nucleoid geometry, and thus, cell morphology. In support of this proposition, many spherical (*e.g.,* see *Figure 1A* in *Kerfeld et al., 2005*) and filamentous (*e.g.,* see *Figure 1* in *Montgomery, 2015*) cyanobacterial cells also show a hexagonal carboxysome arrangement. Furthermore, *S. elongatus* cells grown under environmental conditions that increase carboxysome synthesis also dominantly display hexagonal packing (*e.g.,* see *Figure 2* in *Sun et al., 2016*).

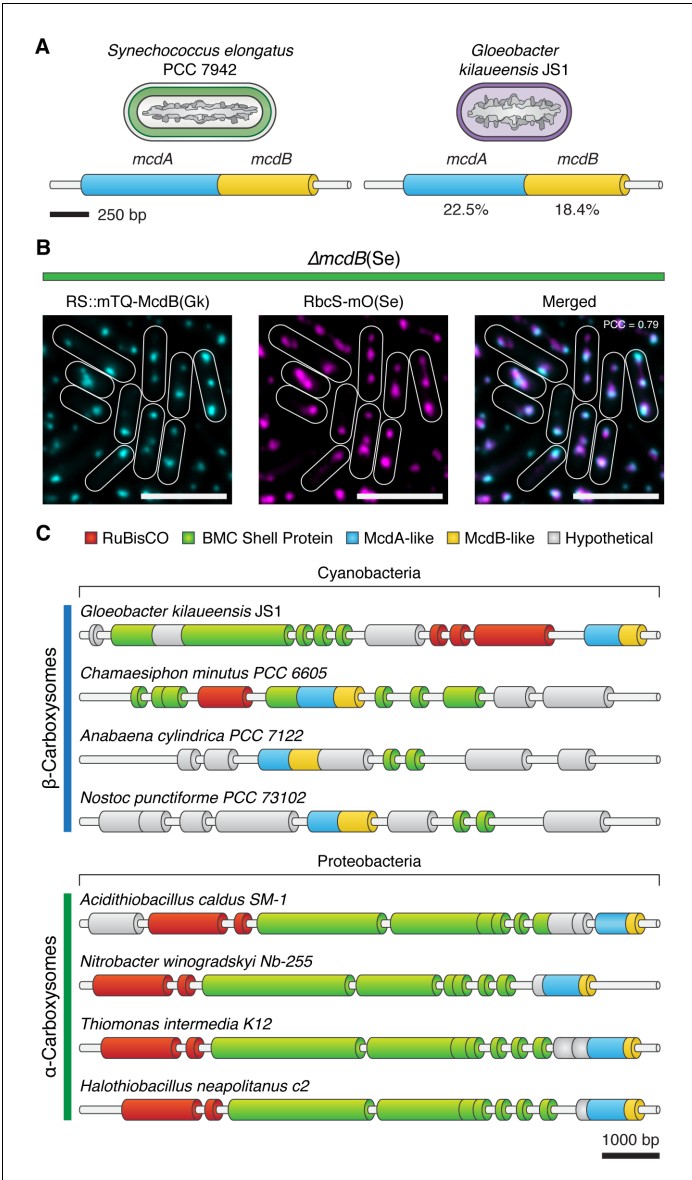

**Figure 8.** Evolutionary conservation of McdAB. (**A**) Cartoon illustration of McdAB operon structures in *S. elongatus* and *Gloeobacter kilaueensis* JS1. (**B**) *Gloeobacter kilaueensis* JS1 McdB colocalizes with *S. elongatus* carboxysomes (RbcS-mO). Scale bar = 5 μm. (**C**) McdA/B-like sequences colocalize in the genome with predicted carboxysome components across diverse microbes.
DOI: https://doi.org/10.7554/eLife.39723.025

## The McdAB system is evolutionarily wide-spread in cyanobacteria

Homologs of ParA-type ATPases have been identified within extended carboxysome operons of cyanobacteria (*Axen et al., 2014*). Therefore, we examined other distant cyanobacterial species for possible McdAB homologs. One such case is the primitive thylakoid-less cyanobacterium *Gloeobacter kilaueensis* JS1, which drives expression of an *mcdA*-like gene from the *rbcL* promoter. Interestingly, upon further examination, we found a small coding sequence following this *mcdA*-like gene with weak similarity to *mcdB*. BlastP determined that *S. elongatus* McdA had 22.5% pairwise sequence identity to the *G. kilaueensis* JS1 McdA-like protein, while *S. elongatus* McdB had only 18.4% pairwise identity to the McdB-like protein (*Figure 8A*). To investigate the possibility that the McdB-like protein of *G. kilaueensis* JS1 functions similarly to *S. elongatus* McdB, we expressed a fluorescent fusion of the *G. kilaueensis* JS1 *mcdB*-like gene, mTQ-McdB(Gk), in our *S. elongatus* Δ*mcdB* strain.

Despite the low primary sequence identity of McdB(Gk), we found that mTQ-McdB(Gk) colocalized with RbcS-mO (PCC = 0.79; n = 248), indicating that mTQ-McdB(Gk) can interact with *S. elongatus* carboxysomes (*Figure 8B*). These results suggest that carboxysome positioning by McdA and McdB may be widespread among cyanobacteria.

## Discussion

### McdA is not cytoskeletal and utilizes the nucleoid to position carboxysomes

Carboxysomes are essential components of the photosynthetic metabolism of cyanobacteria, yet the mechanisms underlying their positioning within the cell has remained an outstanding question. Prior work in *S. elongatus* showed that a ParA-like protein (McdA) was required for maintaining carboxysome positioning (*Savage et al., 2010*). Largely influenced by models for DNA segregation by ParA-type ATPases at the time, the observation that C-terminally tagged McdA (McdA-GFP) oscillated *in vivo* and that carboxysomes were mispositioned following the disruption of McdA or MreB (an actin-related component of the cytoskeleton) led to a widely-adopted hypothesis that carboxysomes were positioned by a cytoskeletal mechanism (*Savage et al., 2010*; *Murat et al., 2010*; *Rae et al., 2013*; *Yokoo et al., 2015*). This model proposed that adjacent carboxysomes were connected via McdA filaments that continually polymerize and depolymerize, exerting physical force upon carboxysomes in order to maintain their equidistant positioning – although no direct evidence for this model has been demonstrated to date.

Here we show that McdA-GFP homogeneously binds non-specifically to DNA, but exhibits no signs of filament formation (*Figure 1A–C* and *Video 1*). We show multiple lines of evidence indicating that McdA is capable of binding DNA in an ATP-dependent manner, and that the bulk of McdA in *S. elongatus* is concentrated upon the nucleoid. Furthermore, we identify a novel protein, McdB, that interacts with McdA, stimulating its intrinsic ATPase activity and release from the nucleoid. Taken together, these results strongly suggest that McdA does not form independently-standing filaments, but instead attaches to the nucleoid body at the center of the cell, using it as a scaffolding surface to support an oscillating wave from one end of the cell to the other. We provide evidence that carboxysomes are in turn tethered to the nucleoid through interactions with McdB, that in turn can bind McdA. Our results provide insight into the molecular mechanisms of McdA, and extend upon the limited characterization in the literature of this unusual ParA-family member. While we confirm that McdA can oscillate *in vivo* as was previously shown (*Savage et al., 2010*), we observe oscillatory waves that traverse the cell within ~10 min, which is significantly faster than that reported for McdA-GFP. The discrepancy may be related to the use of C-terminally tagged McdA reporters, which we find are unable to interact effectively with McdB and was originally over-expressed in a background with an additional endogenous copy of McdA (*Savage et al., 2010*). We observe activities for McdA (tagged on either terminus) that are consistent with other ParA family members, including ATP-dependent DNA binding (*Figure 1A*), self-association (*Figure 2F* and *Figure 2—figure supplement 1A*), and DNA stimulated ATPase activity (*Figure 2H* and *Figure 2—figure supplement 1F*). However, we also observe key distinctions between McdA and more canonical family members (see below), which may be important in the adaption of this system for the segregation of large protein cargos.

Importantly, we do not find any direct evidence supporting filament formation by McdA. Using high-resolution imaging techniques, we observe McdA-GFP to coat evenly across DNA in a carpeted flowcell, without any indication of large oligomer formation. While we cannot exclude the possibility that McdA-GFP fusions are disrupted in their capacity to form filaments, we note that this fusion retains many of its characteristics (*Figure 1A*, *Figure 2—figure supplement 1A*, and *Figure 2—figure supplement 1F*). Furthermore, our data suggests that the morphology of the cyanobacterial nucleoid is important for carboxysome positioning (*Figures 1*, *2JK*, *3C*, *6*, *7*), which could also explain why ΔmreB mutants exhibit disorganized carboxysomes (*Savage et al., 2010*). Both cell morphology and nucleoid topology are grossly altered in ΔmreB strains (*Hu et al., 2007*), suggesting that the influence of MreB in carboxysome positioning is likely indirect. Instead, our results support an alternative model for McdA in carboxysome positioning that does not require an underlying cytoskeleton and which utilizes a Brownian-ratchet based mechanism (see below).

## A unique system for distributing protein-based bacterial organelles

Our model of self-organized carboxysome positioning is both informed by the ParA-based mechanisms used to segregate low-copy number plasmids, but also provides a novel platform to study the dynamics of self-organized protein segregation systems. Low-copy plasmids often contain DNA regions (e.g. *parS*) that bind ParB, which drives the directed and persistent movement of plasmids towards increased concentrations of ParA on the nucleoid (*Vecchiarelli et al., 2010*; *Le Gall et al., 2016*). In this way, it is proposed that ParA can provide a positional cue allowing plasmid cargo to 'surf' along the larger bacterial chromosome without a separate cytoskeletal system (*Vecchiarelli et al., 2012*).

While conceptual similarities exist between the better-established systems for plasmid positioning and results we report here for carboxysome positioning, a number of key distinctions separate McdAB from ParAB models. First, *S. elongatus*' McdA lacks the signature lysine residue in the Walker A box that defines the ParA family of ATPases (<u>K</u>GGXXGT; *Figure 2G*). The serine substitution in McdA at a position universally conserved in ParA members may underlie the unusually high ATPase activity of McdA (*Figure 2G*), which displays a maximum specific activity that is roughly two-orders of magnitude greater than that of other well-studied ParA systems (*Vecchiarelli et al., 2010*; *Ah-Seng et al., 2009*). McdB is an even more divergent protein, bearing no identifiable sequence similarity to any known ParB proteins; indeed, no homologous proteins have been characterized in other species. This novel protein also recognizes and binds a large protein-based cargo (carboxysomes; *Figure 3C,F,H*), further distinguishing it from all characterized ParB-like proteins that recognize genetic cargo. Even though McdB and ParB share no similarity, we find that McdB: (i) interacts with McdA (*Figure 2F*), (ii) stimulates McdA ATPase activity (*Figure 2H*) (iii) removes McdA from DNA (*Figure 2J*), and (iv) is responsible for emergent dynamics of McdA along the nucleoid (*Figure 2D*); analogous to the roles played by ParB in well-characterized plasmid partitioning systems. Furthermore, we observe that a pool of McdB enriched at the carboxysome is necessary to locally deplete McdA (*Figures 4* and *6B–D*), suggesting that prolonged McdB activity may stimulate the local release of McdA from the nucleoid. We propose that McdB is therefore acting to interface carboxysomes with nucleoid-bound McdA, processively pulling this protein cargo towards the highest local McdA concentration, and thereupon stimulating McdA ATPase activity and release (*Figure 6D* and *Videos 3–5*). The parallels between features of the McdAB system and Brownian ratchet ParAB models make it tempting to speculate that McdB has a distinct evolutionary origin from ParB-family members, but that these independent protein families convergently evolved to use nucleoid gradients of ParA-like proteins to segregate entirely different classes of macromolecular structures.

The colocalization of signal between mNG-McdB and carboxysomes (RbcS-mTQ) (*Figure 3C*), coupled with the carboxysome requirement for providing site-specificity to mNG-McdB *in vivo* (*Figure 3H*), provide strong evidence that McdB is associating with carboxysomes and that this interaction is needed for emergent dynamics of McdA (*Figure 3I*). It is curious that McdB is able to associate with a number of different shell proteins in our B2H assay (*Figure 3F*). Taken together with the evidence that a McdB homolog from *G. kilaueensis* JS1 with low sequence-identity still concentrated upon *S. elongatus* carboxysomes (*Figure 8B*), the most parsimonious hypothesis is that McdB-carboxysome shell interactions are mediated by structural and/or charge features common to many distinct shell proteins. Indeed, evolutionarily distant hexameric shell proteins of the bacterial microcompartment (BMC-H) family share a number of similarities in structural features and key residues at hexamer interfaces that are largely conserved (*Cai et al., 2015*; *Sommer et al., 2017*; *Young et al., 2017*). This suggests some of these common structural features could be important in mediating interactions with McdB, which might explain why McdB displays an interaction with different shell protein paralogs. Our B2H analysis also indicates that McdB may have a higher affinity to some shell proteins (CcmK2 and CcmK3) than others (CcmK4, CcmL and CcmO) (*Figure 3F*). This may be related to the observation of clustered carboxysomes in Δ*ccmK3-4* mutants (*Rae et al., 2012*), as this may reduce the amount of McdB recruited to the carboxysome surface. We note however, that given McdB's poor sequence conservation and without further knowledge of the structure and interaction domains of McdB, we cannot rule out that McdB is a 'sticky' protein by the B2H assay and is instead recruited through an alternative adaptor protein to the vicinity of carboxysomes. Moreover, deleting individual shell components, such as CcmK2, CcmL or CcmO, prevents mature

carboxysomes formation and subsequent biogenesis (*Cameron et al., 2013*), preventing *in vivo* testing of McdB/carboxysomes interaction. Additional experiments will be required to identify the domain(s) that mediate McdB-shell interaction, and without a more detailed analysis, it remains possible that McdB can directly integrate within the shell of mature carboxysomes. Some indirect evidence would argue against the possibility that McdB is an integral shell protein, including both our observation that ΔmcdB strains did not possess a high CO2-requiring phenotype and McdB has not been identified in previously-published carboxysome purification studies (*Faulkner et al., 2017*).

## McdA oscillations are a consequence of multiple carboxysomes sharing the same nucleoid in a rod-shaped cell

One surprising result of our study was that we observed that both McdB and carboxysomes themselves were required for the emergence of McdA oscillations along the nucleoid (*Figure 2D*, *3HI*). Furthermore, a critical threshold number of carboxysomes were required to be localized on the same nucleoid in order for McdA oscillation to ensue (generally >3; *Figure 6F*). This suggests that it is not sufficient for McdB to be merely present, it must be specifically localized, concentrated, and/or activated to promote the McdA oscillations. Furthermore, we note that McdA oscillation per se is not required to segregate one carboxysome from another. In cells where we induced the formation of only two carboxysomes, or in cells were the $P_{trc}$ promoter was leaky, these carboxysomes reliably separated from one another despite the fact that no McdA oscillations were present (*Figure 6DF*). Likewise, our Brownian-Rachet simulations were able to recapitulate the separation between carboxysomes *in silico* without any requirement for an oscillating pool of McdA (*Figure 7CDEF*).

These results could suggest that 'global' McdA oscillations have a secondary role in regulating carboxysome positioning relative to the influence of local gradients of McdA on the nucleoid, and also raise other questions related to how McdA oscillations emerge. One possibility is that McdA oscillation itself might be a byproduct of the motion of multiple carboxysomes removing McdA along the nucleoid. Indeed, such a model has been proposed for plasmid segregation, termed 'DNA relay' (*Surovtsev et al., 2016*), where the plasmids recruit ParB to form cargo complexes that themselves oscillate from pole-to-pole in the cell, removing nucleoid-bound ParA in their wake. In the DNA relay model, the long-range motion of the cargo itself drives the emergent oscillation of ParA. Our simultaneous imaging of carboxysomes and McdA dynamics precludes such a model that would require long-range cargo movement (carboxysomes move much shorter distances and over longer time scales than the McdA oscillatory wave), but we cannot rule out more subtle carboxysome motions being involved in the emergence of McdA oscillations. The dynamics of carboxysome motion are complex at rapid time scales and are responsive to McdA wavefronts. During McdA oscillation, we observed that some carboxysomes at the wave front paused, and in some cases, were observed to get sucked into the approaching wave (*Videos 3–5*). While in the wave, carboxysome diffusion was suppressed. As the wave passed, carboxysome diffusion was anisotropic, drifting in the direction of the wave. Away from the wave, carboxysome diffusion was isotropic. An alternative hypothesis is that the global dynamics of McdA oscillation are dependent upon a balanced level of activities between McdA and McdB. In this case, recruitment of soluble McdB to a defined location (the carboxysome) may concurrently act to remove it from the bulk cytosol, reducing the concentration that McdA perceives when not near a carboxysome. There is precedence for this interpretation in the ParA-like family of proteins, including the oscillatory behaviors of MinD and MinE. MinD binds to the plasma membrane when bound to ATP and exhibits an emergent pole-to-pole localization that is driven by the ATPase-stimulating activities of the partner protein MinE (*Lutkenhaus, 2007*). The ratio of these activities is important for their higher-order behaviors and when MinD:MinE ratios become severely unbalanced, oscillatory patterns collapse (*Fange and Elf, 2006*; *Loose et al., 2008*; *Loose et al., 2011*; *Zieske and Schwille, 2014*; *Vecchiarelli et al., 2016*; *MacCready et al., 2017*).

It is intriguing to speculate whether McdA would oscillate in cyanobacteria displaying different morphologies, such as the spherical *Synechocystis sp.* 6803 or the filamentous *Fremyella diplosiphon*. While carboxysomes in these organisms are equidistantly spaced, they display a packing more reminiscent of the hexagonal arrangement rather than a linear distribution (*e.g.*, see *Figure 1A* in *Kerfeld et al., 2005*; *e.g.*, see *Figure 1* in *Montgomery, 2015*). Our modeling suggests that this could be a natural outcome of the McdAB system operating on a nucleoid topology that is more

spherical, rather than rod-shaped. While McdA oscillation could still be possible, it is unclear what patterns would be expected. Further analysis of McdAB dynamics in other cyanobacteria is required to elucidate the effects of nucleoid morphology on McdA pattern formation.

## McdA and McdB may facilitate biogenesis and regulate carboxysome size

Our analysis provides a number of lines of evidence to suggest that McdA and McdB activities can also influence the ultrastructure of carboxysomes in *S. elongatus*. Cyanobacterial strains that are genetic knockouts of *mcdB* display carboxysomes that are significantly enlarged (*Figure 5—figure supplement 1AD*). Furthermore, overexpression of McdB resulted in massive carboxysome globules that sometimes spanned the entire short axis of the cell (*Figure 5—figure supplement 1AG*), while overexpression of McdA resulted in irregularly-shaped carboxysomes with rounded edges (*Figure 5—figure supplement 1AF*). While we cannot rule out the possibility of indirect effects, it is intriguing to speculate that McdA and McdB may act to directly regulate the size or shape of microcompartments as they are formed. In *S. elongatus*, it has been suggested that new carboxysomes bud off from existing carboxysomes (*Cameron et al., 2013*; *Chen et al., 2013*). Our model for carboxysome positioning requires that the interaction of McdB with McdA provide a pulling force exerted on the carboxysome shell that acts to processively move the protein compartment up an McdA gradient. It is therefore possible that these same molecular forces act during the synthesis of a new carboxysome. The relative ratio between McdA and McdB activities may play a role in the differences in carboxysome sizes observed under different environmental conditions or within different species. Future research will be required to confirm the role of McdAB systems in regulating the size of protein-based organelles.

## The McdAB system in other organisms

Carboxysomes exist in two distinct forms, α and β, depending on the form of RuBisCO they encapsulate. While both are found in cyanobacteria, α-carboxysomes also exist in many actinobacteria and proteobacteria. In these organisms, the vast majority of carboxysome-related genes tend to be found at genomic loci near the respective enzymes they encapsulate (*Axen et al., 2014*). We find that *mcdA/B*-like sequences frequently fall in regions near α- and β-carboxysome operons (*Figure 8C*). We propose that the *mcdA/B*-like sequences near the α-carboxysome operon could also function to equidistantly space α-carboxysomes to ensure equal inheritance following cell division. Further study is now needed to determine how widespread the McdAB system is across evolutionary space. Indeed, many BMC classes exist, are widespread in bacteria, and encapsulate a wide array of enzymatic activities beyond Calvin-Benson-Bassham factors (*Axen et al., 2014*; *Kerfeld and Erbilgin, 2015*). While putative McdB homologs are widespread in cyanobacteria and can be identified in many α-carboxysome-containing proteobacteria (*Figure 8C*), it is possible that a more comprehensive bioinformatic approach could identify similar factors associated with other classes of BMC. More broadly, these findings aid in understanding the spatial organization of other protein-based mesoscale assemblies that encode ParA family members and are associated with diverse biological processes, including secretion (*Perez-Cheeks et al., 2012*; *Viollier et al., 2002*), conjugation (*Atmakuri et al., 2007*), chemotaxis (*Thompson et al., 2006*; *Ringgaard et al., 2011*; *Alvarado et al., 2017*), and cell motility (*Youderian et al., 2003*; *Kusumoto et al., 2008*).

## Materials and methods

### Construct designs

All constructs in this study were generated using Gibson Assembly (*Gibson et al., 2009*) from synthetized dsDNA and verified by sequencing. Constructs contained flanking DNA that ranged from 500 to 1500 bp in length upstream and downstream of the targeted insertion site to promote homologous recombination into target genomic loci (*Clerico et al., 2007*).

#### Native fluorescent fusions

For native McdA fluorescent fusions, the fluorescent protein mNeonGreen (mNG) was attached to either the 5' or 3' region of the native *mcdA* coding sequence, separated by a GSGSGS linker. Since

**Table 1.** Cyanobacterial strains used in this study.

| Strain Name | Description/Genotype |
| --- | --- |
| JSM-201 | mNG-McdA |
| JSM-202 | McdA-mNG |
| JSM-203 | mNG-McdB |
| JSM-204 | McdB-mNG |
| JSM-205 | RbcS-mO |
| JSM-206 | mNG-McdA + RbcS-mTQ |
| JSM-207 | mNG-McdB + RbcS-mTQ |
| JSM-208 | mNG-McdAΔparB |
| JSM-209 | mNG-McdAΔ Synpcc7942_1834 |
| JSM-210 | mNG-McdAΔ Synpcc7942_1835 |
| JSM-211 | mNG-McdA + ΔmcdB + RbcS-mO |
| JSM-212 | ΔmcdA + mTQ McdB+RbcS-mO |
| JSM-213 | ΔmcdA + ΔmcdB + RbcS-mO |
| JSM-214 | mNG-McdA + ΔmcdB + RbcS mO+RS::mTQ-McdB |
| JSM-215 | ΔmcdA + mTQ McdB+RbcS mO+RS::mNG-McdA |
| JSM-216 | mNG-McdA + RbcS mTQ + ΔccmK2-ccmO |
| JSM-217 | mNG-McdB + RbcS mTQ + ΔccmK2-ccmO |
| JSM-218 | mNG-McdA + RbcS mTQ+RS:: CcmK2 |
| JSM-219 | mNG-McdA + RbcS mTQ+Ptrc::CcmK2 |
| JSM-220 | RbcS-mTQ + RS::Cdv3 |
| JSM-221 | mNG-McdA + ΔmcdB + RbcS mO+RS::Cdv3 |
| JSM-222 | ΔmcdA + mTQ McdB+RbcS mO+RS::Cdv3 |
| JSM-223 | mNG-McdA + ΔmcdB + RbcS mO+RS::mTQ-McdB(Gk) |

DOI: https://doi.org/10.7554/eLife.39723.026

the upstream coding sequence next to *mcdA* is essential and presumably expressed from the same region of DNA as *mcdA*, the kanamycin resistance cassette was inserted upstream of the *mcdA* promoter to prevent operon disruption, and a duplicate *mcdA* promoter was inserted upstream of kanamycin to drive expression of the essential coding sequence. For the native McdB-mNG construct, mNG was inserted at the 3' end of the *mcdB* coding sequence, separated by a GSGSGS linker, followed by the kanamycin resistance cassette. Alternatively, for the native mNG-McdB construct, the *mcdB* sequence was codon optimized to prevent recombination at this site and mNG was inserted at the 5' end. The kanamycin resistance cassette was inserted downstream. To visualize carboxysomes, a second copy of the *rbcS* promoter and gene, attached at the 3' end with either the fluorescent protein mTurquoise2 (mTQ) or mOrange2 (mO) and separated with a GSGSGS linker, were inserted into neutral site 1.

## Single Deletions

Deletion constructs of plasmid *parB*, *1835*, and *mcdB* were created by replacing the respective coding sequences with a spectinomycin resistance cassette. Likewise, deletion of the carboxysome operon was performed by replacing the entire coding sequence, starting with the *ccmK2* promoter and ending with *ccmO*, with a spectinomycin resistance cassette.

## Native fluorescent fusions with deletion or overexpression

For fluorescent and deletion lines, single plasmids were created that contained 5' fluorescently labeled *mcdA* (mNG) or *mcdB* (mTQ), separated by a GSGSGS linker, that simultaneously deleted *mcdA* or *mcdB* and integrated the *rbcS::rbcS-mO* fluorescent reporter upstream of the *mcdB* coding sequence (*Figure 4ABC*). In these lines, the kanamycin resistance cassette was inserted upstream

similarly to our native mNG-McdA constructs (*Figure 4ABC*). For our Δ*mcdAB* strain, a codon optimized *rbcS-mO* sequence was inserted to replace the entire *mcdA* operon while inserting the kanamycin resistance cassette upstream. Overexpression of mNG-McdA or mTQ-McdB were performed by insertion into neutral site two and expressed using a Ptrc promoter with an attached 5' theophylline riboswitch (*Nakahira et al., 2013*).

### Carboxysome induction systems
To generate a tunable carboxysome operon, we replaced the native *ccmK2* promoter with a Ptrc promoter in the absence of the *lacI* repressor and inserted a theophylline riboswitch (*Nakahira et al., 2013*) on the 5' end of *ccmK2*. Alternatively, we also replaced the *ccmK2* promoter with a Ptrc promoter without a 5' riboswitch on *ccmK2* and inserted the *lacI* repressor upstream. In both constructs, the spectinomycin resistance cassette was inserted upstream of the inserted promoters.

## Culture conditions and transformations
All *S. elongatus* cultures were grown in 125 mL baffled flasks (Corning) containing 50 ml BG-11 medium (SIGMA) buffered with 1 g L$^{-1}$ HEPES to pH 8.3. Flasks were cultured in a Multitron II (atrbiotech.com) incubation system with settings: 80 μmol m$^{-2}$ s$^{-1}$ light intensity, 32°C, 2% $CO_2$, shaking at 130 RPM. Cloning of plasmids was performed in *E. coli* DH5α chemically competent cells (Invitrogen). All *S. elongatus* transformations were performed as previously described (*Clerico et al., 2007*). Cells were plated on BG-11 agar with either 12.5 mg ml$^{-1}$ kanamycin or 25 mg ml$^{-1}$ spectinomycin. Single colonies were picked into 96-well plates containing 300 μl of BG-11 with identical antibiotic concentrations. Cultures were verified for complete insertion via PCR and removed from antibiotics.

## Bacterial-two-hybrid analysis
N- and C-terminal T18 and T25 fusions of McdA, McdB and shell proteins CcmK2, CcmK3, CcmK4, CcmL, CcmO and CcmP were constructed using plasmid pKT25, pKNT25, pUT18C and pUT18, sequence-verified and co-transformed into *E. coli* BTH101 in all pairwise combinations (*Karimova et al., 1998*). Several colonies of T18/T25 cotransformants were isolated and grown in LB medium with 100 μg/ml ampicillin, 50 μg/ml kanamycin and 0.5 mM IPTG overnight at 30°C with 225 rpm shaking. Due to the self-assembling nature of carboxysome shell proteins, overnight IPTG induction for cotransformants bearing T18/T25 shell protein fusions was carried out at 0.1 mM IPTG. Overnight cultures were spotted on indicator MacConkey plates supplemented with 100 μg/ml ampicillin, 50 μg/ml kanamycin and 0.5 mM IPTG. Plates were incubated at 30°C up to 48 hr before imaging.

## Induction strains
Overproduction of mNG-McdA and mTQ-McdB were accomplished by inducing strains with 1500 μM theophylline for 48 hr. For carboxysome induction under the riboswitch, strains were incubated in 400 μM theophylline (one carboxysome) or 600 μM theophylline (two carboxysomes) for 24 hr prior to imaging. Alternatively, for carboxysome induction under the Ptrc promoter and LacI repressor, cells were incubated with 1000 μM IPTG for 16 hr prior to imaging. To increase cell lengths of the carboxysome reporter only strain or the Δ*mcdA*, Δ*mcdB* and Δ*mcdAB* with the carboxysome reporter strains, RS::Cdv3 was overexpressed for 48 hours using 1500 μM theophylline.

## Fluorescence microscopy
All live-cell microscopy was performed using exponentially growing cells. Two mL of culture was spun down at 5000xg for 30 s, resuspended in 200 μl of BG-11 and 2 μl transferred to a square 1.5% agarose +BG-11 pad on glass slides. All images were captured using a Zeiss Axio Observer A1 microscope (100x, 1.46NA) with an Axiocam 503 mono camera except the carboxysome induction experiments. Carboxysome induction experiments were performed using a Nikon Ti2-E motorized inverted microscope with LED-based light sources (100x, 1.45NA) with a Photometrics Prime 95B Back-illuminated sCMOS Camera. Image analysis was performed using Fiji v 1.0.

## Transmission Electron Microscopy

Cultures were grown to OD750 = 0.7 in BG-11. Cells were pelleted and fixed overnight at 4 ˚C with 2.5% formaldehyde/2.5% glutaraldehyde in 0.1 M sodium cacodylate buffer (pH 7.4), suspended into a 2% agarose bead and cut into ~1 mm cubes. Following three washes with 0.1 M sodium cacodylate buffer, cells were suspended in 1% osmium tetroxide/1.5% potassium ferrocyanide and incubated overnight at 4 ˚C. After incubation, cells were washed with HPLC-quality $H_2O$ until clear. Cells were then suspended in 1% uranyl acetate and microwaved for 2 min using a MS-9000 Laboratory Microwave Oven (Electron Microscopy Science), decanted, and washed until clear. Cells were dehydrated in increasing acetone series (microwave 2 min) and then embedded in Spurr's resin (25% increments for 10 min each at 25°C). A final overnight incubation at room temperature in Spurr's resin was done, then cells were embedded in blocks which were polymerized by incubation at 60 ˚C for three days. Thin sections of approximately 50 nm were obtained using an MYX ultramicrotome (RMC Products), post-stained with 1% uranyl acetate and Reynolds lead citrate, and visualized on a JEM 100CX II transmission electron microscope (JEOL) equipped with an Orius SC200-830 CCD camera (Gatan).

## MicrobeJ quantification

Cultures were grown to $OD_{750}$ = 0.7 in BG-11. Multiple individual images of the fluorescent reporters and chlorophyll autofluorescence were obtained for each strain and analyzed using MicrobeJ 5.11 n. In each line, cell perimeter detection was performed using the rod-shaped descriptor and default thresholding algorithm. Carboxysome detection was performed using the foci function with a tolerance of 15 and Z-score of 50. For fluorescent McdA lines, localization was quantified via detection of the single brightest point (tolerance = 2000). For fluorescent McdB lines, localization was quantified via multiple smoothed foci detections (tolerance = 15 and Z-score = 50). Associations, shape descriptors, profiles and distances were recorded for each strain. Heatmaps were automatically generated with counts, contour and a spot size of 5. Mean foci area and standard deviation for each maxima was automatically calculated.

## McdA-GFP-6xHis expression and purification

The gene sequence, *mcdA–GFP–6xHis*, was codon optimized for *E. coli* and synthesized by Genscript. The fragment was inserted into the NcoI/BamHI cloning sites of the expression vector pET15b to create the pAV30 plasmid. pAV30 was transformed into BL21 (AI) cells (Invitrogen) and a 100 mL overnight culture containing 100 μg/mL of carbenicillin was grown at 20°C with shaking at 225 rpm. LB supplemented with 100 μg/mL of carbenicillin and a drop of Antifoam Emulsion (1 L per 2.5 L Fernbach flask ×4) was pre-warmed to 37°C and inoculated with 10 mL of overnight culture per flask. The cells were grown at 37°C with shaking at 225 rpm to an O.D. of 0.4. The flasks were then plunged in an ice bath until the temperature of the culture dropped to 16°C. Protein expression was then induced at O.D. 0.6 by the addition of 10 mL of a 0.1 M IPTG/20% Arabinose solution to each flask. Cells were then grown overnight at 16°C with shaking at 225 rpm (~15 hr induction). The cells were transferred to 1 L Beckmann bags and bottles, which were spun in a JLA 8.1 rotor at 4,500 rpm for 1 hr. The supernatant was poured out, and the cell pellets were frozen in the bags with liquid nitrogen and stored at −80°C. Frozen cell pellets were combined in a beaker with 10 mL of cold Lysis Buffer per gram of cell pellet (~150 mL total), three Protease Inhibitor Mixture Tablets (Sigma) and 0.1 mg/mL lysozyme (Sigma). A homogenizer was used to ensure that the cell pellets were thoroughly dispersed, and two passes through a Microfluidizer lysed the cells. The lysate was cleared with a 30 min ultracentrifugation at 35,000 rpm and 4°C using a 45Ti rotor. The lysate was then passed through a 0.45 μm syringe filter. Using a peristaltic pump, the cleared lysate (~200 mL) was loaded at a flow rate of 2 mL/min onto a 5 mL HisTRAP HP cassette (GE) and equilibrated with Lysis Buffer (50 mM HEPES–KOH (pH 7.6), 1 M KCl, 10% Glycerol, 20 mM Imidazole (pH 7.4), 2 mM β-mercaptoethanol). The protein was eluted with a 20 mM to 1 M imidazole gradient (total volume = 60 mL). Peak protein fractions were pooled and concentrated using an Amicon Ultra Centrifugal Device (10 KD MWCO). The sample was passed through a 26/10 salt-exchange column equilibrated in Q-Buffer (50 mM HEPES–KOH (pH 7.5), 200 mM KCl, 10% Glycerol, 0.1 mM EDTA, 2 mM DTT). The sample was then immediately loaded onto a 1 mL Mono Q 5/50 anion exchange column (GE) equilibrated in Q-Buffer. The protein was eluted with a 200 mM to 1 M KCl gradient. Peak

fractions were pooled and concentrated to a no more than 100 µM. The sample was then separated over a 10/300 GL Superdex200 gel-filtration column equilibrated in Q Buffer (but with 600 mM KCl). Peak fractions were pooled, concentrated to no more than 100 µM, frozen with liquid nitrogen, and stored at −80°C.

## 6xHis-MBP-McdA expression and purification

Due to insolubility issues encountered when expressing McdA-6xHis, a construct was designed where a 6xHis-MBP-tag was encoded upstream of a Tobacco Etch Virus (TEV) cleavage site and fused to the N-terminus of the mcdA gene in a pET15b expression backbone to create pAH2 plasmid. pAH2 was transformed into ArcticExpress (DE3) competent cells (Agilent) and protein expression was carried out by growing transformants at 37°C and 225 rpm until an OD600 of 0.6–0.8 was reached. Following an ice bath plunge to lower the culture temperature to 15°C, protein expression was induced with the addition of 0.5 mM IPTG. Induction was allowed to continue overnight at 15°C. The cells were pelleted, flash frozen with liquid nitrogen, and stored at −80°C. Cells were then lysed in Buffer A (50 mM HEPES pH 7.6, 50 mM KCl, 10% glycerol, 20 mM imidazole pH 7.4, 5 mM BME, 50 µg/ml lysozyme, 1.25 kU benzonase, 2 Protease Inhibitor Cocktail tablets) using a probe sonicator with 15 s on, 15 s off pulsation for 8 min. Cell debris was removed by centrifugation at 14,000 rpm for 40 min in a FiberliteTM F15−8 × 50 cy Fixed Angle Rotor (ThermoFisher Scientific) and the resulting lysate was filtered through a 0.45 µm syringe filter prior to being loaded onto a HiTrapTM Q HP 5 ml cassette (GE) connected in tandem to a 5 ml HiTrapTM TALON Crude cassette (GE). The protein was eluted from the Q cassette with a 50 mM – 1 M KCl gradient in an anion exchange chromatography step. The His-tagged protein was then eluted from the TALON column with a 20 mM – 1M imidazole gradient. Peak fractions were pooled, concentrated and further separated by gel filtration on a Superdex200 10/300 GL column (GE) pre-equilibrated with 50 mM HEPES pH 7.6, 50 mM KCl, 10% glycerol, 5 mM DTT. Individual peak fractions were concentrated to no higher than 20 µM and frozen aliquots were kept at −80°C.

## McdB-6xHis expression and purification

A codon-optimized gene sequence of *mcdB-6xHis* was inserted into the NcoI/BamHI cloning sites of the expression vector pET15b, yielding pAV42 plasmid. The construct was transformed into BL21(AI) and protein expression was carried out in the same manner as McdA-GFP-His. Frozen cell pellets were thawed and resuspended in lysis buffer containing 50 mM HEPES pH 7.6, 500 mM KCl, 10% glycerol, 20 mM imidazole pH 7.4, 5 mM MgCl$_2$, 2 mM BME, 50 µg/ml lysozyme, 1.25 kU benzonase, 2 Protease Inhibitor Cocktail tablets. Resulting cell lysate was centrifuged, filtered and loaded onto a 5 ml His-Trap$^{TM}$ Ni-NTA cassette (GE). Following protein elution with a 20 mM – 1M imidazole gradient, peak fractions were pooled, concentrated and loaded onto a Superdex200 HiLoad 16/600 PG column pre-equilibrated with 50 mM HEPES pH 7.6, 500 mM KCl, 10% glycerol, 5 mM MgCl$_2$, 2 mM DTT for final separation. Peak fractions were concentrated to no more than 70 µM and flash frozen aliquots were kept at −80°C.

## ATPase assay

ATPase assays were performed in a buffer containing 50 mM HEPES (pH 7.6), 10 mM MgCl$_2$, 100 mM KCl, 0.1 mg/ml BSA, 2 mM DTT, and 0.1 mg/ml sonicated salmon sperm DNA (when present). Unlabeled ATP was spiked with [γ-$^{32}$P]-ATP and purified from contaminating $^{32}$P$_i$ prior to use with a 1 ml gel filtration (P-2 fine resin, Bio-Rad) column. The radiolabeled ATP mix was added to reactions at 1 mM. Reactions were assembled on ice at the protein concentrations indicated, with His-MBP-McdA, McdA-GFP-His, F SopA-His or P1 ParA being added last. The 20 µl reactions were incubated for 1 hr at 30°C and immediately quenched by adding 10 µl of a 1% SDS, 20 mM EDTA solution. Two microliters of the quenched reactions were spotted and analyzed by thin-layer chromatography as previously described (*Fung et al., 2001*). Due to the feeble ATPase activities of SopA-His and P1 ParA, specific activities were determined from experiments carried out as shown above, but the 30°C incubation period was carried out for 3 hr.

## DNA binding assay

Electrophoretic mobility shift assays (EMSAs) were performed in a final reaction volume of 10 µl in a buffer containing 50 mM HEPES (pH 7.6), 5 mM $MgCl_2$, and 100 mM KCl with 10 nM pUC19 plasmid (2.8 kb) as the DNA substrate. At the concentrations indicated, McdA-GFP-His or His-MBP-McdA was incubated for 30 min at 23°C with ADP, ATP or ATPγS (1 mM). When used, McdB-His was added at the concentrations specified. Reactions were then mixed with 1 µl 80% glycerol, run on 1% agarose gel in 1X TAE at 110V for 45 min and stained with ethidium bromide for imaging. The peak fractions representing the dimer form of His-MBP-McdA from Superdex200 size exclusion chromatography were used.

## TIRFM of McdA-GFP binding to a DNA-carpeted flowcell

Quartz flowcell construction and DNA-carpeting of the flowcell surface were performed as previously described (*Vecchiarelli et al., 2014*). For the imaging of McdA-GFP-His binding to the DNA carpet, prism-type TIRFM was performed using an Eclipse TE2000E microscope (Nikon) with a PlanApo $60 \times$ NA = 1.40 oil-immersed objective and magnifier setting at $1.5\times$. Movies were acquired using an Andor DU-897E camera (Andor Technology) with integrated shutter. The camera settings were digitizer, 3 MHz (14-bit gray scale); preamplifier gain, 5.2; vertical shift speed, 2 MHz; vertical clock range: normal, electron-multiplying gain 40, EM CCD temperature set at –98°C, baseline clamp ON, exposure time 100 ms, frame rate 0.5 Hz. The baseline of ~100 camera units was subtracted from the intensity data. The excitation for McdA-GFP-His was provided by 488 nm diode-pumped solid-state (Sapphire, Coherent) laser. Total internal reflection fluorescence illumination had a Gaussian shape in the field of view with measured horizontal and vertical half maximum widths of ~65 µm $\times$ 172 µm at 488 nm. Intensity data for the DNA carpet-bound populations of McdA-GFP were taken from the middle of the illumination profile. The laser power of 488 nm illumination was 15 µW. Metamorph seven software (Molecular Devices) was used for camera control and image acquisition. ImageJ was used for analysis. The display brightness and contrast were set to the same levels for all TIRFM movies. ImageJ was used for conversion of Metamorph movies (.stk) into. avi format and Adobe Premiere was used for added text. Movie accelerations are indicated in the movie and figure legends.

McdA-GFP (0.5 µM) was preincubated in McdA Buffer [50 mM Hepes (pH 7.6), 100 mM KCl, 10% (vol/vol) glycerol, 5 mM $MgCl_2$, 2 mM DTT, 0.1 mg/mL α-casein, 0.6 mg/mL ascorbic acid] with 1 mM of the indicated nucleotide (or no nucleotide). The sample was incubated for 15 min in a 1 ml syringe connected to one of the two inlets of a Y-shaped flowcell. The sample was infused onto the DNA carpet at a rate of 20 µL/min. The fluorescence intensity of McdA-GFP that bound the DNA carpet was measured over time. At t = 3 min, flow from the sample inlet was stopped and immediately switched to the second inlet that was connected to a wash buffer (McdA Buffer without McdA-GFP or nucleotide). Wash buffer was flowed at a rate of 20 µL/min, and the decrease in fluorescence intensity was monitored over time. The two-inlet flowcell had a Y-patterned configuration and imaging took place at the point of flow convergence to minimize the effect of protein rebinding to the DNA carpet during dissociation when flow was switched to the wash buffer.

## Theoretical model and computational method

We find that McdA, McdB, and the carboxysome cargo show *in vivo* dynamics strikingly similar to that found for ParA-mediated DNA partition systems. Therefore, we leveraged our established Brownian ratchet model of ParA/ParB-mediated partition (*Hu et al., 2015*; *Hu et al., 2017*) to theoretically interrogate the carboxysome positioning process in cyanobacteria.

Briefly, the model describes the mechanochemical interplay between nucleoid-bound McdA and carboxysome-bound McdB. McdA and McdB in the current model fulfil exactly the same roles of ParA and ParB as in the low-copy plasmid partition case, respectively. While carboxysome alone diffuses randomly, its motility can be greatly modulated when carboxysome-bound McdB interacts with nucleoid-bound McdA. Specifically, carboxysome-bound McdB stimulates the ATPase activity of nucleoid-bound McdA, which triggers the dissociation of McdA from the nucleoid substrate surface. The slow rate of dissociated McdA resetting its DNA-binding capability generates an McdA-depleted zone behind the moving cargo. The resulting asymmetric McdA distribution perpetuates the directed movement of the carboxysome cargo. Transient tethering arising from the McdA-McdB

contacts collectively drives forward movement of the cargo and also quenches diffusive motion in orthogonal directions. This way, McdA/McdB interaction – when at proper mechanochemical coupling – drives directed and persistent movement of carboxysomes (*Hu et al., 2015*; *Hu et al., 2017*).

The model treats carboxysomes as circular disks that move on the nucleoid surface, which is modeled as a 2D simulation domain. The simulation domain is bounded by the reflective boundary condition. To study the effects of nucleoid geometry on carboxysome positioning, we constructed the simulation domains to mimic I.) a circular nucleoid and II.) a more elongated rounded rectangle nucleoid, which consists of a rectangle with two spherical caps at the ends of its long axis.

We simulated the model by the same kinetic Monte Carlo technique as in (*Hu et al., 2015*; *Hu et al., 2017*), which describes the coupling between the stochastic reaction-diffusion processes involving McdA and McdB, and their mechanochemical interplay. Specifically, we investigated the effects of carboxysome number and nucleoid geometry on carboxysome positioning. In each case defined by different carboxysome number and nucleoid geometry, the simulation starts with the initial positions of the carboxysomes that cluster around the center of the simulation domain (see *Figure 7CE* in the main text). For each case, we identify the parameter regime in which the carboxysomes undergo 'directed segregation' — a motility mode in which the cargoes move away from each other and then become relatively stationary (*e.g.*, see *Figure 3* in *Hu et al., 2017*). That is, the carboxysomes are segregated and then stably positioned with a large inter-spacing. We then tracked the time evolutions of each of the simulated trajectories that are 10 min long, from which the final positions of, and separation distances between, carboxysomes were then calculated and reported in each case (average ± standard deviation, n = 36 trajectories).

## Acknowledgements

We would like to thank the labs of Dr. Matthew Chapman and Dr. Kiyoshi Mizuuchi labs for use of equipment and reagents, as well as Dr. Cheryl Kerfeld and Dr. Beronda Montgomery for helpful conversations and suggestions in preparing this manuscript. The pET15b expression vector used for His-MBP fusion to McdA was a kind gift from Dr. Maria Schumacher. Purified P1 ParA and F SopA-His were kind gifts from the Funnell and Mizuuchi labs respectively. This work was supported by the National Science Foundation (Award Numbers 1517241; to KWO and DCD and 1817478; to AGV) and by research initiation funds to AGV provided by the MCDB Department, University of Michigan. Additional institutional and equipment support was provided by Office of Basic Energy Sciences, Office of Science, US Department of Energy (DE–FG02–91ER20021; to DCD).

## Additional information

### Funding

| Funder | Grant reference number | Author |
| --- | --- | --- |
| National Science Foundation | 1517241 | Katherine W Osteryoung Daniel C Ducat |
| National Science Foundation | 1817478 | Anthony G Vecchiarelli |
| Basic Energy Sciences | DE-FG02-91ER20021 | Daniel C Ducat |

The funders had no role in study design, data collection and interpretation, or the decision to submit the work for publication.

### Author contributions

Joshua S MacCready, Conceptualization, Data curation, Formal analysis, Validation, Investigation, Visualization, Methodology, Writing—original draft, Conceived the project, Designed experiments, Performed all experiments; Pusparanee Hakim, Data curation, Formal analysis, Validation, Investigation, Methodology, Designed experiments, Performed all experiments; Eric J Young, Data curation, Formal analysis, Validation, Visualization, Performed all experiments; Longhua Hu, Data curation, Software, Formal analysis, Validation, Methodology, Performed all experiments, Designed and

carried out computer simulations; Jian Liu, Data curation, Software, Formal analysis, Supervision, Validation, Methodology, Designed and carried out computer simulations; Katherine W Osteryoung, Funding acquisition, Writing—original draft; Anthony G Vecchiarelli, Conceptualization, Resources, Data curation, Formal analysis, Supervision, Funding acquisition, Validation, Investigation, Methodology, Writing—original draft, Conceived the project, Designed experiments, Performed all experiments; Daniel C Ducat, Conceptualization, Supervision, Funding acquisition, Methodology, Writing—original draft, Conceived the project, Designed experiments

### Author ORCIDs
Joshua S MacCready (iD) https://orcid.org/0000-0001-6438-8110
Pusparanee Hakim (iD) https://orcid.org/0000-0002-9018-8179
Eric J Young (iD) https://orcid.org/0000-0002-6770-6310
Katherine W Osteryoung (iD) http://orcid.org/0000-0002-0028-2509
Anthony G Vecchiarelli (iD) https://orcid.org/0000-0002-6198-3245
Daniel C Ducat (iD) https://orcid.org/0000-0002-1520-0588

### Decision letter and Author response
Decision letter https://doi.org/10.7554/eLife.39723.031
Author response https://doi.org/10.7554/eLife.39723.032

## Additional files

### Supplementary files
• Transparent reporting form
DOI: https://doi.org/10.7554/eLife.39723.027

### Data availability
All data generated or analysed during this study are included in the manuscript and supporting files. Source data files have been provided for Figure 1—figure supplement 1 and Figure 2—figure supplement 1.

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
