## [Decision Letter]

Thank you for submitting your article "Protein Gradients on the Nucleoid Position the Carbon-fixing Organelles of Cyanobacteria" for consideration by *eLife*. Your article has been reviewed by two peer reviewers, and the evaluation has been overseen by a Reviewing Editor and Gisela Storz as the Senior Editor. The following individual involved in review of your submission has agreed to reveal their identity: Emilia MF Mauriello (Reviewer #2).

The reviewers have discussed the reviews with one another and the Reviewing Editor has drafted this decision to help you prepare a revised submission.

The manuscript "Protein Gradients on the Nucleoid Position the Carbon-fixing Organelles of Cyanobacteria" by MacCready et al. addresses the mechanism by which carboxysomes are positioned and segregated in their model organism *Synechococcus elongates*. Particularly, the authors present a detailed study that addresses some important unanswered questions about the mechanism responsible for proper segregation of carboxysomes. They identify a previously uncharacterized protein McdB, which together with its partner protein, the ParA-like ATPase McdA, is required for carboxysome segregation. McdB regulates McdA ATPase activity and the oscillatory behavior of McdA over the nucleoid. Furthermore, their data suggest that McdB is a component of the carboxysome and in this way provides a link between McdA function and carboxysome positioning. Based on their microscopy data and mathematical modelling the authors propose a Brownian ratchet-based mechanism for McdA/McdB function.

There are however three major points that need to be addressed before the paper is published:

1) Provide statistical analyses of the imaging (for which they must have all the raw data);

2) Provide evidence for the functionality of their fusion;

3) Provide an additional simple in vitro experiment of DNA binding.

1) Quantification of fluorescence microscopy experiments. The manuscript is highly based on microscopy analysis of protein localization, however, many microscopy experiments are not quantified or presented with statistical analysis. Some experiments are only based on the analysis of a single cell. Proper quantification and statistical analysis is needed. In particular such analyses should address the following points:

– Figure 1A: Demographic analysis. How many cells show asymmetric localization of McdA?

– Figure 1C. Have these small mNG-McdA clusters been seen in multiple cells? Have they ever been colocalized with the carboxysomes? Finally, why mNG-McdA would colocalize with the carboxysomes? We would expect it to immediately dissociate upon ATP hydrolysis.

– Figure 1—figure supplement 1D: Quantification of RbcS-mTQ localization in wild-type and mNG-McdA backgrounds. What is normal localization of RbcS?

– How common are the dim McdA foci forming in the wake of McdA oscillation? Quantification is needed. Is this a common phenomenon?

– Demographic quantification of microscopy data of mNG-McdA in *ΔparB*, *Δ1835* and *Δ1834* in Figure 2B-D is needed. The images do seem to support the author's conclusion but proper quantification is needed (e.g. by demographs of mNG-McdA signal along the cell length).

– In Figure 2I, proper quantification of McdB localization is needed. Does the number of McdB foci increase with increasing cell length? Are McdB foci regularly distributed similar to RbcS? Analysis of McdB signal as a function of cell length will tell this.

– Furthermore, in Figure 2K, proper quantification of mNG-McdB and RbcS-mTQ signal is needed in order to show co-localization of the two proteins in the cell. This is very important, since it is a major conclusion of the manuscript that McdB integrates into the carboxysomes. Proper quantification of McdB recovery is also needed.

– The authors conclude on the recovery of McdB in a single cell that McdB association with carboxysomes is stable (Supplementary Figure 1L). The average recovery rate of McdB from multiple cells should be performed. Furthermore, photoactivation of e.g. PAmCherry-McdB would be the correct experiment to determine of McdB is stably associated with carboxysomes – this experiment would allow analysis of release of McdB from carboxysomes and hence the stability of its association with them.

– There is no quantification of RbcS-mTQ, mNG-McdB and mNG-McdA localization in absence of carboxysomes. The McdA signal does not appear uniform like McdB and RbcS. There is no quantification of mNG-McdA signal in respect to RbcS-mTQ signal in Figure 6B and C. Again the conclusion is based on the single cell presented in the figure. Proper analysis of mNG-McdA signal in respect to RbcS-mTQ signal is needed – how general is this observation?

– Again, in regard to Figure 6D statements as "we frequently observed" is used but no data for frequency is given – how common is this behavior of McdA?

2) Functionality and integrity of the different protein fusions used in the study: mNG-McdA (vs. McdA-mNG); mNG-McdB (vs. McdB-mNG); Rbcs-mO; RbcsTQ needs to be illustrated by western blots and phenotypic assays. It would be also nice to clearly state the reasons beyond the choice of one tag over another one in the different experiments (McdA-GFP-His over McdA-mNG, for example). This is critically important because a major conclusion of the manuscript that McdA does not form filaments but instead functions via a Brownian ratchet-based mechanism. This is a new model for McdA function, which previous data suggest function via a filament forming-mechanism. However, one of the primary experiments for this conclusion is based on in vitro DNA binding assays using purified McdA-GFP-His. However, there is no experiment testing if this fusion is functional *in vivo* – though the authors do show that it binds DNA. This is particularly concerning since the McdA-mNG fusion is non-functional, which the authors clearly show in Figure 1. It is likely that polymers of McdA-GFP-His are not observed because GFP-His blocks oligomerization of McdA-GFP-His and consequently rendering it non-functional, similar to the McdA-mNG fusion. The functionality of McdA-GFP-His *in vivo* needs to be tested. In case it is non-functional, then *in vitro* experiments ought to be performed with a fusion that is functional *in vivo*.

3) It would be an important addition to this work to show that the McdA binding to DNA (+/- ATP) is affected by the presence of McdB in both gel shifts and flowcell. Have the authors tried this experiment? From the two hybrid system results, I expect that the McdA-GFP-McdB binding is compromised: could the authors use His-MBP-McdA, which has been shown to hydrolyse ATP in the presence of McdB and DNA, for this experiments instead than McdA-GFP? The authors could not purify mNG-McdA nor McdA because they are insoluble. Have they tried to purify it in the presence of ATP/ADP? Does McbB bind DNA?

Other comments that would help improve the manuscript:

– Figure 4C. The authors describe the carboxysomes observed in this figure irregular and randomly distributed, but it is hard to discriminate the difference between these carboxysomes and those observed in wild type. I can even catch some of the hexagonal distribution described later in the manuscript. Anyway, the carboxysomes distribution in the *ΔmcdA ΔmcdB* double mutant is very different from that observed in single mutants. This suggests that the McdA and McdB might also have some independent functions. Could the authors comment on that?

– It would be nice if the fluorescence micrographs in Figure 6B-D were comparable with those in Figure 6F. In Figure 6B-D we can nicely observe the relocalization of McdA between two carboxysomes and the movement of the one carboxysome towards high McdA concentrations. It would be nice to show the appearance of mNG-McdA oscillations upon the addition of IPTG in the strain bearing *lacI*, in time lapse experiments. The authors should be able to show it even in the absence of a microfluidic device. Conversely, in Figure 6B-D it would be nice to have larger fields showing multiple cells like in Figure 6F.

– The authors conclude that a pool of McdB enriched at the carboxysome is necessary to locally deplete McdA signal (Figure 4 and Figure 6B-D), suggesting that prolonged McdB activity may stimulate the local release of McdA from the nucleoid. It would significantly improve the author's conclusion if they identified an McdB variant that still localizes to the carboxysome but do not interact with McdA. If the author's conclusion is correct, then McdA should be uniformly distributed on the nucleoid in this McdB background while the McdB variant still colocalized to the carboxysomes. It is a major assumption in their modelling that McdB regulates McdA behavior in such a way.

Edits:

– Subsection “McdA Dynamically Patterns Along the Nucleoid”, first sentence of last paragraph: Replace McdA with McdA-GFP.

– The authors could comment on the fact that the shift pattern observed on Figure 1F is typical of ParA (Soj) proteins nucleating on DNA.

– Figure 1G. What is the 500 bp DNA sequence used in the flowcell?

– Subsection “Carboxysome Positioning is Disrupted in McdA/McdB Mutants”: “In contrast to other strains that lack a McdA oscillation (see below), we did not observe depleted McdA signal on the nucleoid in the vicinity of carboxysomes within a ∆*mcdB* background”. This sentence is not clear to me. Have foci of non-oscillating mNG-McdA been observed near the carboxysomes elsewhere in the manuscript?

– Figure 5. Do the authors have the DAPI staining for Figures 5A-F. I was wondering what is nucleoid shape in the observed strains. In TEM, have clustered carboxysomes ever been observed in wild type?

– Figure 1—figure supplement 1A-B and Figure 1—figure supplement 1F-K are redundant and can be removed.

– Figure 1—figure supplement 1D could become a main figure.

---

## [Author Response]

[…] There are however three major points that need to be addressed before the paper is published:1) Provide statistical analyses of the imaging (for which they must have all the raw data);2) Provide evidence for the functionality of their fusion;3) Provide an additional simple in vitro experiment of DNA binding.1) Quantification of fluorescence microscopy experiments. The manuscript is highly based on microscopy analysis of protein localization, however, many microscopy experiments are not quantified or presented with statistical analysis. Some experiments are only based on the analysis of a single cell. Proper quantification and statistical analysis is needed. In particular such analyses should address the following points:– Figure 1A: Demographic analysis. How many cells show asymmetric localization of McdA?

We apologize for not emphasizing the fact that the dynamics described here using our native fluorescent fusions of McdA were essentially fully penetrant in a cell population (≥99% of cells). We have provided several lines of statistical analyses in the main text and supplementary emphasizing this point. Specifically, we have added the following statements to the main text:

“Interestingly, our native C-terminally tagged reporter (McdA-mNG) did not show dynamic oscillations and instead formed a uniform distribution of signal along the longitudinal axis (≥99% of cells; n = 950 cells) (Figure 1D and Figure 1—figure supplement 1A). Alternatively, a N-terminally tagged reporter (mNG-McdA) displayed robust oscillations (≥99% of cells; n = 442 cells) (periodicity of 15.3 min per 3.3 µm, ~5-6x faster than previously reported using McdA-GFP (Savage et al., 2010)) that formed a bimodal distribution of signal intensity (Figure 1E,G, Figure 1—figure supplement 1B and Video 2).”

Similar quantifying statements are found throughout the Results section.

We have also provided Video 2 showing mNG-McdA oscillations occurring in greater than 99% of cells in the field of view (n = > 500 cells).

Further, we have added supplementary figure panels that display MicrobeJ heatmap statistical analyses of the fluorescence signals associated with mNG-McdA, which shows asymmetric localization and McdA-mNG, which does not oscillate (Figure 1—figure supplement 1). The results from these statistical treatments are consistent with our interpretation of the data as described in the main text. We have also added heatmap statistical analyses of the fluorescence signals associated with a number of other reporter constructs (Figure 3—figure supplement 1).

– Figure 1C. Have these small mNG-McdA clusters been seen in multiple cells? Have they ever been colocalized with the carboxysomes? Finally, why mNG-McdA would colocalize with the carboxysomes? We would expect it to immediately dissociate upon ATP hydrolysis.

The reviewer recognizes a valid point. The signal is weak and is certainly a transient phenomenon that only occurs during the brief time period in which mNG-McdA is traversing the long axis of the cell. As such, examples are few. Therefore, we have decided to remove this image from the main text as we cannot be sure that the signal is due to interaction with carboxysomes, or perhaps an interaction with regions of the nucleoid with higher DNA densities. Also, the finding is cursory and does not significantly add to the conclusions of the paper.

Despite removing this image, we would like to respond to the reviewer’s query as to why we would expect mNG-McdA to transiently colocalize with carboxysomes. ParA-mediated positioning has largely been studied in the context of moving genetic material; plasmids in particular. The ParB protein from plasmid P1 forms punctate foci in vivo that remain colocalized with plasmids bearing *parS* (Erdmann et al., 1999; Sengupta et al., 2010). P1 ParA uniformly distributes over the nucleoid region of the cell, and also forms foci that colocalize with relatively immobile plasmids bearing *parS* (Hatano and Niki, 2010). But during periods of plasmid movement, the colocalized ParA foci disappear and only reappear once the plasmid has been repositioned. Chromosomal ParAs also have been observed as two discrete populations in vivo. Both ParA from *Caulobacter crescentus* and ParAI from *Vibrio cholerae* form foci that colocalize with their ParB-*parS* complexes during periods of relative immobility (Fogel and Waldor, 2006; Ptacin, 2010). In both cases, ParA disappears from the mobile ParB-*parS* complex during segregation, and the nucleoid-bound population of ParA redistributes in response to ParB-*parS* motion. Therefore, the transient mNG-McdA association with McdB-bound carboxysomes we observed is consistent with that found for other better studied ParA-type ATPases.

In the context of carboxysomes, as the reviewer points out, McdB interaction should result in McdA release from the nucleoid but this only occurs after first physically interacting. Indeed the counteracting roles of AB association versus ATPase stimulated release of McdA is a critical feature of our diffusion-ratchet model (Hu et al., 2015, 2017). Directed and persistent positioning requires a sufficient number of McdA⋅ATP–McdB bonds to transiently tether the carboxysome cargo and quench diffusion while still being able to dissociate rapidly enough to drive carboxysome movement. The proper coordination between the timescales of the mechanical action and the chemistry of the McdAATP–McdB bonds orchestrates the synchrony between bond tethering and dissociation in driving persistent cargo positioning.

– Figure 1—figure supplement 1D: Quantification of RbcS-mTQ localization in wild-type and mNG-McdA backgrounds. What is normal localization of RbcS?

Quantification RbcS-mOrange localization in wild-type cells is already present as a MicrobeJ heatmap in Figure 5A. The distribution of carboxysomes is essentially identical to the heat map we have now provided for RbcS-mTQ localization in mNG-McdA (Figure 1—figure supplement 1D) and mNG-McdB contexts (Figure 3—figure supplement 1C).

– How common are the dim McdA foci forming in the wake of McdA oscillation? Quantification is needed. Is this a common phenomenon?

See above.

– Demographic quantification of microscopy data of mNG-McdA in ΔparB, Δ1835 and Δ1834 in Figure 2B-D is needed. The images do seem to support the author's conclusion but proper quantification is needed (e.g. by demographs of mNG-McdA signal along the cell length).

We now provide demographic quantification of mNG-McdA in all strains requested:

– “However, deletion of pANL *parB* did not disrupt oscillation of mNG-McdA (≥99% of cells; n = 554 cells)”;

– “…deletion of *Synpcc7942_1835* had no observable effect on mNG-McdA oscillation (≥99% of cells; n = 834 cells)”;

– “…deletion of *Synpcc7942_1834* resulted in complete loss of mNG-McdA dynamics (≥99% of cells; n = 373 cells)”.

We also provide population-scale demographs of mNG-McdA signal along the cell length for all mutants as requested (see Figure 1—figure supplement 1E, F, G).

– In Figure 2I, proper quantification of McdB localization is needed. Does the number of McdB foci increase with increasing cell length? Are McdB foci regularly distributed similar to RbcS? Analysis of McdB signal as a function of cell length will tell this.– Furthermore, in Figure 2K, proper quantification of mNG-McdB and RbcS-mTQ signal is needed in order to show co-localization of the two proteins in the cell. This is very important, since it is a major conclusion of the manuscript that McdB integrates into the carboxysomes. Proper quantification of McdB recovery is also needed.

We now provide demographic quantification of the fluorescent signals of McdB-mNG and mNG-McdB in a cell population:

– “C-terminal fusions of McdB displayed a diffuse localization with random punctate-like patterns (≥99% of cells; n = 371 cells)”;

– “In contrast, N-terminal mNG-McdB was observed as multiple discrete fluorescent foci near the central longitudinal axis of the cell (≥99% of cells; n = 699 cells), a result that strongly resembles the localization pattern of native carboxysomes”.

Also, we now provide a Pearson’s Correlation Coefficient to highlight the strong colocalization of mNG-McdB and RbsC-mTQ (i.e. carboxysomes) throughout the cell population:

“Both the mNG-McdB and RbcS-mTQ signals strongly colocalized (≥99% of cells; PCC= 0.92; n = 316 cells) as fluorescent foci near the long central axis of the cell”.

We believe the PCC value of 0.92 is a strong addition to the paper, and precludes an analysis of McdB signal as a function of cell length – the data will be identical to that shown in Figure 1—figure supplement 1C.

– The authors conclude on the recovery of McdB in a single cell that McdB association with carboxysomes is stable (Supplementary Figure 1L). The average recovery rate of McdB from multiple cells should be performed. Furthermore, photoactivation of e.g. PAmCherry-McdB would be the correct experiment to determine of McdB is stably associated with carboxysomes – this experiment would allow analysis of release of McdB from carboxysomes and hence the stability of its association with them.

We agree with the reviewer that photoactivation would be a more ideal experiment to address the stability of McdB on carboxysomes. Because of a number of problems with quantifying McdB signal in the context of a photosynthetic cell, we have decided to remove our current FRAP quantification. First, because there are many photosynthetic pigments in cyanobacteria, the autofluorescence is high – especially in the red wavelengths. This precludes the use of pAmCherry and many other photoactivable dyes/fusions. Secondly, when performing FRAP, if the laser power is too low, fluorescence in the ROI does not significantly decline. If it’s too high, the autofluorescence significantly increases, due to saturation of the light harvesting machinery in the electron transport chain and a corresponding increase in the quantum yield of chlorophyll fluorescence (Campbell et al., 1998). Finally, because recovery is on the scale of minutes (which is also on the scale of McdA oscillations) carboxysomes frequently move during the recovery period. For example, non-irradiated carboxysomes often diffuse into the FRAP ROI, or irradiated carboxysomes move within the Z-plane. Therefore, quantification across multiple cells is currently difficult. Since we are only able to provide representative data, and this result is not directly related to the central conclusions of this paper, we have removed the FRAP data and its associated text.

There is no quantification of RbcS-mTQ, mNG-McdB and mNG-McdA localization in absence of carboxysomes. The McdA signal does not appear uniform like McdB and RbcS.

We now provide quantification of RbcS-mTQ, mNG-McdB and mNG-McdA localization in absence of carboxysomes in a cell population:

– “We therefore examined the localization of mNG-McdB in a *ΔccmK2LMNO* background and found that mNG-McdB signal was diffuse (≥99% of cells; n = 389 cells) in the absence of carboxysomes…”;

– “Interestingly, in the absence of carboxysomes, mNG-McdA did not oscillate and formed a homogenous distribution along the nucleoid similar to that of our mNG-McdA*∆mcdB* strain (≥99% of cells; n = 227 cells)”;

– “In the absence of theophylline, we observed that RbcS-mTQ signal was diffuse and mNG-McdA was distributed homogenously along the nucleoid (≥99% of cells; n = 204 cells) (Figure 5—figure supplement 1H), consistent with our prior results of from *∆ccmK2LMNO* mutants (Figure 3G).”

We also now provide population-scale demographs of RbcS-mTQ, mNG-McdB and mNG-McdA localization in absence of carboxysomes along the cell length as requested (see Figure 3—figure supplement 1EF).

– There is no quantification of RbcS-mTQ, mNG-McdB and mNG-McdA localization in absence of carboxysomes. The McdA signal does not appear uniform like McdB and RbcS. There is no quantification of mNG-McdA signal in respect to RbcS-mTQ signal in Figure 6B and C. Again the conclusion is based on the single cell presented in the figure. Proper analysis of mNG-McdA signal in respect to RbcS-mTQ signal is needed – how general is this observation?– Again, in regard to Figure 6D statements as "we frequently observed" is used but no data for frequency is given – how common is this behavior of McdA?

We now provide a PCC value for Figure 6B and C.

“In either case, mNG-McdA signal was highly reduced in areas of RbcS-mTQ signal (≥99% of cells; PCC = 0.20; n = 391 cells), indicating McdB-bound carboxysomes have mNG-McdA depleted in their vicinity.”

2) Functionality and integrity of the different protein fusions used in the study: mNG-McdA (vs. McdA-mNG); mNG-McdB (vs. McdB-mNG); Rbcs-mO; RbcsTQ needs to be illustrated by western blots and phenotypic assays.

Functionality of different protein fusions is addressed via a variety of distinct approaches in the current manuscript. With regard to McdA and McdB, we show that the reporter fusions are sufficient to recapitulate all of the known functions of these proteins, to date. Because all of our reporter lines are the result of “knock-ins” to the endogenous loci, there is no other pool of untagged proteins that could rescue functions. For example, the endogenous mNG-McdA fusion oscillates within the cell and is sufficient for carboxysome positioning (Figures 1E-H, Figure 2B,C, Figure 1—figure supplement 1B,D-F). Likewise, endogenous mNG-McdB localizes to carboxysomes and is sufficient to drive McdA emergent oscillations, and is sufficient for ordered carboxysome positioning in the absence of any additional pool of untagged McdB in the cell (Figure 3C, Figure 3—figure supplement 1B,C). We go to lengths to show that the C-terminal fusions that have different localization patterns in vivoalso do not compliment the known functions of McdA and McdB and also behave differently in some in vitro experiments: for McdA (see Figure 2F and Figure 2—figure supplement 1A,F), for McdB (see Figure 2F and Figure 2—figure supplement 1A). With regard to the functionality of RbcS-fluorophore fusions, we cannot tag the endogenous copy of the protein. Moreover, RbcS antibodies that recognize *S. elongatus* RbcS are not commercially available. However, we note that the reaction carried out by RuBisCO is essential for growth in *S. elongatus*, which is a strict photoautotroph. If RbcS-fusions acted as a dominant negative mutant, this construct would be lethal. Additionally, if RbcS-fusions compromised the integrity of the carboxysome, a high CO_2_-requiring phenotype would result, as intact carboxysomes are an essential component of the carbon concentrating mechanism (e.g., Cameron et al., 2013). We do not observe a high CO_2_-requiring phenotype in any of our lines aside from the carboxysome deletion strain.

The authors are somewhat uncertain what is meant by the reviewer’s request to show “integrity” of the various protein fusions. We interpret this statement to mean that the fluorophore reporters are not partially degraded in a way that separates the fluorophore from the tagged protein. However, if this were the case, the results of the striking localization patterns (i.e., oscillating waves, ordered puncta) would be highly unlikely – if separated from their tagged protein, fluorophores should adopt a diffuse localization pattern (or at worst, be concentrated to polar inclusion bodies). We note that even in the cases of the C-terminally tagged McdA and McdB which we suspect interferes with McdA-McdB interactions (for which we provide multiple in vitro that suggest impaired functionality – Figure 2—figure supplement 1A, Figure 2—figure supplement 1F, and Figure 3—figure supplement 1D), the fluorescent signal for these expressed proteins is not diffuse. McdA-mNG concentrates in the cell near the nucleoid (Figure 1D), and McdB-mNG displays an irregular punctate pattern (Figure 3A). Carboxysomes have been routinely visualized by tagging components of the shell and/or RuBisCO subunits and the localization patterns we observe for RbcS are consistent with other carboxysome reporter lines (e.g., Savage et al., 2010, Cameron et al., 2013, Sun et al., 2016, Niederhuber et al., 2017). We note that commercially-available antibodies are not available for many of the proteins in this study, precluding an exhaustive Western analysis.

However, a commercial antibody is available for mNeonGreen, therefore we present a Western blot (Author response image 1) of increasing concentrations of purified mNG (~27 kD), and mNG-McdB (~44 kD), mNG-McdA (~54 kD) and mNG-Ftn2 (~98 kD control) cell lysate. We do not observe any degradation bands in the blot, consistent with maintenance of fusion integrity. A non-specific cross-reactive band (~50kD) found in all cyanobacterial sample lysates runs near the predicted size of mNG-McdA.

It would be also nice to clearly state the reasons beyond the choice of one tag over another one in the different experiments (McdA-GFP-His over McdA-mNG, for example). This is critically important because a major conclusion of the manuscript that McdA does not form filaments but instead functions via a Brownian ratchet-based mechanism.

We indeed used a number of McdA fusions and we agree with the reviewer that the explanations as to why we used them when we did was not completely clear in the previous version of the manuscript. In the current version, we have made a number of changes in the text both in the Results and Discussion explaining why a particular fusion of McdA was used. Briefly, we would like to highlight that McdA-GFP was initially chosen because this was the fluorescent fusion first used by the Savage et al. Science paper that first unveiled the genetic requirement for a ParA-type ATPase for carboxysome positioning. A cytoskeletal model was proposed without any direct evidence for a such a filament-based mechanism. We moved on to N-terminal fusions after finding McdA-GFP was as functional as mNG-McdA in a number of assays that we show here (in vivo oscillation, B2H with McdB, ATPase assays). To clarify this distinction in the main manuscript we have modified text in a few sections – for example we have added the following:

– “Since a C-terminal GFP fusion of McdA was previously used to observe McdA oscillation and its involvement in carboxysome positioning in vivo (Savage et al., 2010), we purified McdA-GFP-His and examined its capacity to bind DNA via an Electrophoretic Mobility Shift Assay (EMSA).”

– “We then sought to examine endogenous localization and dynamics of McdA, therefore we generated N- and C-terminal fluorescent fusions of mNeonGreen (mNG) to McdA using the native mcdA promoter and chromosomal location. […] Alternatively, a N-terminally tagged reporter (mNG-McdA) displayed robust oscillations (≥99% of cells; n = 442 cells) (periodicity of 15.3 min per 3.3 µm, ~5-6x faster than previously reported using McdA-GFP (Savage et al., 2010)) that formed a bimodal distribution of signal intensity (Figure 1E,G, Figure 1—figure supplement 1B and Video 2).”

– “Since an N-terminal fusion of McdA was more functional in vivo, we performed an EMSA with an N-terminal fusion of McdA. […] Together, our results demonstrate that ATP-bound N- or C-terminally tagged McdA binds DNA and McdA-GFP-His does not display indications of polymer formation at the resolution limits of our microscope.”

Additionally, we chose mNG for our in vivo analysis over the original GFP for microscopy reasons. First, fluorescent signal from many GFP-related proteins are too weak in cyanobacterial cells due to native background autofluorescence from their photosynthetic pigments. Indeed, the original Savage et al. paper overexpressed McdA-GFP using a constitutive promoter, thereby bypassing this issue. However, ParA/B stoichiometry is incredibly important for their emergent dynamics and since we were tagging the native genes and expressing them using their native promoter, we chose mNG because it is currently the brightest fluorescent protein, is yellowshifted so we can avoid autofluorescence imaging under YFP and is highly monomeric to prevent unintended self-interaction.

This is a new model for McdA function, which previous data suggest function via a filament forming-mechanism. However, one of the primary experiments for this conclusion is based on in vitro DNA binding assays using purified McdA-GFP-His. However, there is no experiment testing if this fusion is functional in vivo – though the authors do show that it binds DNA. This is particularly concerning since the McdA-mNG fusion is non-functional, which the authors clearly show in Figure 1. It is likely that polymers of McdA-GFP-His are not observed because GFP-His blocks oligomerization of McdA-GFP-His and consequently rendering it non-functional, similar to the McdA-mNG fusion. The functionality of McdA-GFP-His in vivo needs to be tested. In case it is non-functional, then in vitro experiments ought to be performed with a fusion that is functional in vivo.

We recognize the importance of using functional reporter proteins and it is because of these concerns that we have taken pains to characterize multiple different reporter fusions in this work (e.g., C-terminal vs. N-terminal), and to express reporter fusions off of their endogenous promoters, where possible. In responding to this critique about the C-terminal fusion of McdA, it is important to stress that there is currently no direct evidence in support of the current cytoskeletal model for McdA-based positioning of carboxysomes. Savage et al.used only a C-terminal GFP fusion of McdA in their experiments, and for this reason, we start this manuscript by showing in a number of assays that McdA-GFP, to which the cytoskeletal model is predicated (without direct data) is not fully functional in its association with McdB. Since there is no direct data for a filament model here or in the Savage paper, the suggestion that McdA-GFP is not functioning as a polymer due to blocking an oligomerization domain, which there is also no evidence for among all ParAs studied, is an assertion that the polymer model is true because it has not yet been proven false. Formally, speaking some higher order oligomer of McdA (higher than a dimer) may still be possible, but again there is currently no evidence towards this model. We would also like to highlight that this is an ongoing debate in the Min and Par fields. Additionally, we have made a number of attempts to purify a variant of McdA that is tagged on the N-terminus with a fluorophore, and these proteins were unfortunately insoluble or prone to degradation in vitro (see below). Therefore, because we cannot repeat the TIRF experiments using an N-terminal McdA variant, we have restructured portions of our manuscript (especially see Results, subsection “McdA Dynamically Patterns Along the Nucleoid”, and Discussion, subsection “McdA is Not Cytoskeletal and Utilizes the Nucleoid to Position Carboxysomes”) to better highlight the context of the literature and the lack of direct evidence for the filament model.

3) It would be an important addition to this work to show that the McdA binding to DNA (+/- ATP) is affected by the presence of McdB in both gel shifts and flowcell. Have the authors tried this experiment? From the two hybrid system results, I expect that the McdA-GFP-McdB binding is compromised: could the authors use His-MBP-McdA, which has been shown to hydrolyse ATP in the presence of McdB and DNA, for this experiments instead than McdA-GFP? Have they tried to purify it in the presence of ATP/ADP? Does McbB bind DNA?

We agree with the reviewer and performed all requested experiments. We now show that N-terminally fused His-MBP-McdA can shift nsDNA in an EMSA in the presence of ATP (Figure 1I and Figure 2I). We also performed an McdB titration with a fixed concentration of His-MBP-McdA to show that increasing concentrations of McdB resolves this shift (Figure 2J). We also performed an McdB titration without McdA to show that McdB does not bind nsDNA directly (Figure 2K). We thank the reviewer for the suggestion as it has significantly strengthened the conclusions of this paper that the nucleoid plays a key role in the positioning of carboxysomes via the McdAB system.

The authors could not purify mNG-McdA nor McdA because they are insoluble.

McdA shows poor solubility despite the addition of hydrotropic agents such as ATP or ADP. We have also tried purifying McdA using a number of expression constructs and expression strains in *E. coli*. We also attempted purification straight from *S. elongatus* instead of *E. coli*. Our construct for in vivo imaging (His-mNG-GSGSGS-McdA) was highly prone to cleavage during multiple purification attempts. We changed the GSGSGS linker to a TEV cleavage site, which helped resolve the cleavage issue slightly, however protein yield dropped significantly. We will continue to search for a purification protocol for McdA either from *S. elongatus* or from another cyanobacterial species harboring the McdAB system. But for the purposes of introducing the McdAB system, the McdA fusions used here support all expected functions of a ParA-type ATPase – in vivo oscillation, self-association, partner-stimulated ATPase activity and DNA release in vitro.

Other comments that would help improve the manuscript:– Figure 4C. The authors describe the carboxysomes observed in this figure irregular and randomly distributed, but it is hard to discriminate the difference between these carboxysomes and those observed in wild type. I can even catch some of the hexagonal distribution described later in the manuscript. Anyway, the carboxysomes distribution in the ΔmcdA ΔmcdB double mutant is very different from that observed in single mutants. This suggests that the McdA and McdB might also have some independent functions. Could the authors comment on that?

We agree with the reviewer, and for the reason stated it was critical for us to perform the Microbe J analysis of carboxysome distribution, and carboxysome foci size in a population of cells for each mutant line (see Figure 5A-G).

“Carboxysome distribution displayed some variation even in wildtype strains, therefore we quantified carboxysome distributions in hundreds of cells utilizing MicrobeJ to automatically detect and characterize fluorescent carboxysome foci.”

To address the reviewers query regarding the McdAB mutant, carboxysomes are indeed irregular and randomly distributed at a population level. We would like to note that carboxysomes were significantly weaker in fluorescence intensity in the McdAB double mutant. In order to see carboxysomes in Figure 4C, the illumination intensity was increased 4-fold, and this partially accounts for the increased background signal observed in these strains. We speculate that the decreased RbcS-mO signal in the double mutant is due to a change in the promoter genomic context, and/or reduced transcriptional read-through (see construct schematics, Figure 4A-C and subsection “Carboxysome Positioning is Disrupted in McdA/McdB Mutants”, second paragraph). We do suggest in the paper that McdAB may also be involved in the regulation of carboxysome biogenesis, size, and shape. These are potential functions we look forward to addressing in a future work.

– It would be nice if the fluorescence micrographs in Figure 6B-D were comparable with those in Figure 6F. In Figure 6B-D we can nicely observe the relocalization of McdA between two carboxysomes and the movement of the one carboxysome towards high McdA concentrations. It would be nice to show the appearance of mNG-McdA oscillations upon the addition of IPTG in the strain bearing lacI, in time lapse experiments. The authors should be able to show it even in the absence of a microfluidic device. Conversely, in Figure 6B-D it would be nice to have larger fields showing multiple cells like in Figure 6F.

We agree with the reviewer(s) that capturing the emergence of mNG-McdA oscillation upon induction of the carboxysome operon would be a nice addition to our manuscript. However, this type of experiment would need to be performed over 12 hours (Cameron et al., 2010) and we currently lack the appropriate equipment for prolonged cyanobacterial growth on our microscope (2% CO_2_ and a light source capable of minimal elimination in the photosynthetically active range).

– The authors conclude that a pool of McdB enriched at the carboxysome is necessary to locally deplete McdA signal (Figure 4 and Figure 6B-D), suggesting that prolonged McdB activity may stimulate the local release of McdA from the nucleoid. It would significantly improve the author's conclusion if they identified an McdB variant that still localizes to the carboxysome but do not interact with McdA. If the author's conclusion is correct, then McdA should be uniformly distributed on the nucleoid in this McdB background while the McdB variant still colocalized to the carboxysomes. It is a major assumption in their modelling that McdB regulates McdA behavior in such a way.

The new data included in Figure 2J significantly strengthens our claim that McdB regulates McdA attachment to nucleoid DNA. In these gel shift assays, we show that McdB leads to the release of McdA from DNA in a concentration-dependent manner. While we agree that a McdB mutant like that mentioned by the reviewer could allow us to test this in vivo, we regard identification of such a mutant as beyond the scope of this work. This is especially the case because McdB is a novel protein with no structural information or precedence.

Current efforts to identify McdB diversity across cyanobacteria has revealed that McdB is largely a disordered protein with minimal conservation from one species to another (see Author response image 2). Given this, it is not yet obvious which domains/residues would be important for McdB/McdA and McdB/Carboxysome interaction. Therefore, we agree with the reviewer(s) that this effort is important for understanding the system, but currently with this type of analysis it is not yet obvious which residues are critical and will be the focus of future study.

**Author response image 2. respfig2:** 

Edits:– Subsection “McdA Dynamically Patterns Along the Nucleoid”, first sentence of last paragraph: Replace McdA with McdA-GFP.

Corrected.

– The authors could comment on the fact that the shift pattern observed on Figure 1F is typical of ParA (Soj) proteins nucleating on DNA.

The titration of McdA-GFP and His-MBP-McdA show a fairly linear shift of DNA with an increased concentration of ATPase. Therefore, a nucleation event is not apparent in these EMSAs alone. Also, the slowed mobility of DNA in an EMSA, on its own, has little information in regards to the nature or mode in which the protein is binding DNA. Also, our McdA-GFP experiments on a DNA carpet, show slow and homogeneous binding, with no evidence towards a nucleation activity. Therefore, we would not like to overly speculate as to the nature in which McdA is binding nsDNA in our EMSAs. The nature of McdA binding to nsDNA will certainly be addressed further in a future study.

– Figure 1G. What is the 500 bp DNA sequence used in the flowcell?

We have added to the text that the source of nsDNA is Sonicated Salmon Sperm DNA.

– Subsection “Carboxysome Positioning is Disrupted in McdA/McdB Mutants”: “In contrast to other strains that lack a McdA oscillation (see below), we did not observe depleted McdA signal on the nucleoid in the vicinity of carboxysomes within a ∆mcdB background”. This sentence is not clear to me. Have foci of non-oscillating mNG-McdA been observed near the carboxysomes elsewhere in the manuscript?

We agree that this sentence was confusing. It has been removed in the current version of the manuscript.

– Figure 5. Do the authors have the DAPI staining for Figures 5A-F. I was wondering what is nucleoid shape in the observed strains.

We do not have the requested images. We are looking towards other methods of nucleoid labelling, especially those that can be used to non-destructively label DNA in living cells (i.e., fluorescent fusions of nucleoid associated proteins in *S. elongatus*) so as to address the very interesting question of how nucleoid structure may play a role in McdAB-mediated trafficking of carboxysomes.

In TEM, have clustered carboxysomes ever been observed in wild type?

Yes, by TEM imaging, 2 carboxysomes can occasionally be found in proximity to one another (see Figure 4B). However, it is rare to see more than 2 carboxysomes clustered in wild type TEM. Because TEM is not a high-throughput imaging technique and is limited to visualizing structures only within the same cross-sectional plane, we do not attempt to quantify the clustering with this method. It is possible that the instances of 2 carboxysomes near one another in WT are transient events and/or related to biogenesis of new carboxysomes. By contrast, it is extremely common to visualize 2 or more carboxysomes clustering in ∆*mcdA*, ∆*mcdB*, or ∆*mcdAB* cross-sections.

– Figure 1—figure supplement 1A-B and Figure 1—figure supplement 1F-K are redundant and can be removed.

These panels depicting fields with many cells were originally included to show that the phenotypes reported were not isolated/rare events. From many of the points raised by the reviewers regarding population statistics when using fluorescent variants of McdA, we believe these figures are important. To increase their impact, they have been modified with the addition of MicrobeJ heatmaps of the fluorescent signals of mNG-McdA or McdA-mNG in the cell line indicated.

– Figure 1—figure supplement 1D could become a main figure.

We agree and have added the merged image as Figure 1F. We have also kept Figure 1—figure supplement 1D if the reader is interested in viewing the unmerged mNG-McdA and RbcS-mTQ channels independently.

References

Campbell, D1., Hurry, V., Clarke, AK., Gustafsson, P., and Oquist, G. (1998). Chlorophyll fluorescence analysis of cyanobacterial photosynthesis and acclimation. *Microbiol. Mol. Biol. Rev*. 62(3):667-83.

Erdmann, N., Petroff., T and Funnell, BE. (1999). Intracellular localization of P1 ParB protein depends on ParA and parS. *Proc. Natl. Acad. Sci.* 96(26): 14905–14910.

Fogel, MA., and Waldor, MK. (2006). A dynamic, mitotic-like mechanism for bacterial chromosome segregation. *Genes Dev*. 20(23):3269-82.

Hu, L., Vecchiarelli, AG., Mizuuchi, K., Neuman, KC., and Liu, J. (2015). Directed and persistent movement arises from mechanochemistry of the ParA/ParB system. *Proc. Natl. Acad. Sci*. 112(51):E7055-64.

Niederhuber, MJ., Lambert, TJ., Yapp, C., Silver, PA., and Polka, JK. (2017). Superresolution microscopy of the β-carboxysome reveals a homogeneous matrix. *Mol. Biol. Cell*. 28(20):2734-2745.

Ptacin, JL., and Shapiro, L. (2010). Initiating bacterial mitosis: understanding the mechanism of ParA-mediated chromosome segregation. *Cell Cycle*. 9(20):4033-4.